# Characterization and Hydrological Analysis of the Guarumales Deep-Seated Landslide in the Tropical Ecuadorian Andes

**Alexandra Urgilez Vinueza** [1,*] **, Jessica Robles** [2] **, Mark Bakker** [1] **, Pablo Guzman** [2,3] **and Thom Bogaard** [1]

[1] Water Management Department, Delft University of Technology, 2628 CD Delft, The Netherlands; Mark.Bakker@tudelft.nl (M.B.); T.A.Bogaard@tudelft.nl (T.B.)

[2] Hidropaute Business Unit, Corporación Eléctrica del Ecuador (CELEC EP), Cuenca 43R8+29, Azuay, Ecuador; jessica.robles@celec.gob.ec (J.R.); pguzman@uazuay.edu.ec (P.G.)

[3] Environmental Engineering School, Faculty of Science and Technology, Universidad del Azuay, Cuenca 010107, Azuay, Ecuador

\* Correspondence: a.r.urgilezvinueza@tudelft.nl

**Abstract:** The high landslide risk potential along the steep hillslopes of the Eastern Andes in Ecuador provides challenges for hazard mitigation, especially in areas with hydropower dams and reservoirs. The objective of this study was to characterize, understand, and quantify the mechanisms driving the motions of the Guarumales landslide. This 1.5 km$^2$ deep-seated, slow-moving landslide is actively moving and threatening the "Paute Integral" hydroelectric complex. Building on a long time series of measurements of surface displacement, precipitation, and groundwater level fluctuations, we analyzed the role of predisposing conditions and triggering factors on the stability of the landslide. We performed an analysis of the time series of measured groundwater levels and drainage data using transfer functions. The geological interpretation of the landslide was further revised based on twelve new drillings. This demonstrated a locally complex system of colluvium deposits overlying a schist bedrock, reaching up to 100 m. The measured displacement rates were nearly constant at ~50 mm/year over the 18 years of study. However, the measurement accuracy and time resolution were too small to identify possible acceleration or deceleration phases in response to hydro-meteorological forcing. The groundwater and slope drainage data showed a lagged response to rainfall. Finally, we developed a conceptual model of the Guarumales landslide, which we hope will improve our understanding of the many other deep-seated landslides present in the Eastern Andes.

**Keywords:** deep-seated landslide; time series analysis; geodetical monitoring; hydrogeological classification; Ecuadorian Andes

## 1. Introduction

Landslides are the movement of earth down the slope of a hill or mountain. Gravity is the primary driver of this movement, but water and other factors, including anthropogenic influences, may play a role as well [1]. There are several classification systems that describe the characteristics of the movement, including timing and predisposing and triggering factors [1–5]. Apart from the morphological classifications of landslides, Brönnimann [6] proposed a classification system where the hydrogeological 'architecture' and predisposing conditions for landslide occurrence are defined by the permeability and degree of saturation of the slope layers. This approach allows researchers to assess the hydrogeological predisposing conditions and triggering mechanisms behind specific slope instabilities, and to characterize different landslides under the same hydrogeological conditions.

The occurrence of landslides is affected by time-dependent factors. Effective stress in landslides is controlled by the water content within the slope and is dynamically linked to slope deformation and progressive failure [7]. In several studies, rainfall is identified as the threshold parameter that determines the occurrence of landslides (e.g., Guzzetti et al., [8]), but other factors must be considered as well, including antecedent water content, time-variant geotechnical parameters within the regolith, vegetation cover and land use, and the tectonic activities around the area (e.g., [1,9–13]).

Deep-seated landslides require water to accumulate in the landslide body to initiate movement (either due to rainfall or snowmelt), unless they are triggered by a seismic event. Acceleration can vary from days to several weeks after a meteorological event has happened and the hydrological threshold is reached [1] (defined as the condition after which an acceleration of the slope occurs), when seismicity is not the trigger. Groundwater variation can be linked to the occurrence of deep-seated landslides, but other factors also affect movement, including the different geological compositions of the regolith and bedrock, and the permeability and quality of bedrock (e.g., fractures) [11].

Landslides in tropical areas have been relatively under-studied compared to other areas (e.g., Europe, USA). Underlying factors that are essential for landslide occurrence are permanently present in tropical mountainous areas, including rainfall throughout the entire year [14] (present in 50% of the areas around the tropical belt, including Ecuador), tectonic activity, and erosion processes that influence slope geometry and surface conditions leading to changes in predisposing factors. Additionally, tropical areas are mainly located in developing countries, where anthropogenic factors such as human settlements, agriculture, change of land use, and mining are not always adequately planned and controlled, which may become important factors to consider for landslide occurrence [15].

Located in South America, Ecuador is crossed by the Andes mountain range and is situated in a geo-dynamically active region. The Ecuadorian Andes is formed by elevated heights (up to ~6000 m; e.g., Chimborazo volcano). Metamorphosed Triassic and Jurassic plutons dominate, separated by screens of metamorphosed sedimentary and volcanic rocks. The most important structure is the sub-Andean Fault (major reverse fault), and the Inter-Andean Valley with Oligocene and Miocene ignimbrites, that obscures the western limit of metamorphic rocks that form the Cordillera Real [16]. The presence of faults increases seismic risk, leading to deformation processes of the terrain (e.g., [17,18]), and aiding the formative phases of the Ecuadorian Andes [17].

Some key reported landslides in the Ecuadorian Andes (i.e., with major economical and societal repercussions) include the rotational movements of Paccha in 2004, Guasuntos in 2000, La Josefina in 1993, and debris avalanches in Las Moras in 1985 [19]. Several detailed case studies of landslides in the Andes showed the importance of land cover and land cover conversion, e.g., from forest to pasture, on slope deformation in south eastern Ecuador [20,21]. Among the predisposing factors that influence landslide occurrence, the most common are lithological and hydrogeological conformation [22], geomorphology [23], volcanic activity, land use and land use change [20,24], and anthropogenic impacts [25], while triggering factors include rainfall and seismic activity (e.g., [26–28]).

At the transition between the Andes and the low-lying Amazon rainforest, landslides can be further enhanced by the construction of artificial lakes created for hydropower production, due to an increased pore water pressure at the toe of the slope [29]. Approximately 35% of Ecuador's electricity is generated from three dams, which form the Paute Integral hydroelectric complex, over the Paute River. Along two of those reservoirs and their vicinities, researchers identified twenty-one deep-seated landslides [30]. These landslides pose a critical risk to the communities, the operation of the lakes, and the surrounding infrastructure.

Guarumales is one of the twenty-one identified landslides located in the Paute Integral hydroelectric complex and is classified as a deep-seated landslide. Various studies have been conducted on the Guarumales landslide (e.g., [31–34]). However, detailed research on the long-term hydrological factors that may affect the landslide dynamics was not performed prior to this study.

The objectives of this study were to characterize, understand, and quantify the possible driving mechanisms underlying the Guarumales landslide. Special attention was paid to predisposing

conditions, such as geology/lithology, and the relationship between the slope acceleration (if any observed), and possible triggering factors, such as rainfall and subsequent groundwater fluctuations. A detailed understanding of the Guarumales deep-seated, slow-moving landslide is expected to shed light on the other landslides around the Paute Integral hydroelectric complex, which will contribute to a better hazard and risk assessment for the entire region.

## 2. Description of the Study Area

The Guarumales landslide (~2°35′ S, 78°30′ W) is a ~1.5 km$^2$ deep-seated landslide located in the Eastern Andes (Cordillera Real), 110 km from the city of Cuenca. The altitude at the Paute River basin varies from ~400 to ~4600 m.a.s.l. [32]. The geology of Ecuador is highly complex since the Andes have grown by a combination of events like compression, uplift, intrusion, crustal thickening, and volcanism. In Ecuador, the boundary between accreted terranes and S American continental crust is not clear [35]. However, some models have been proposed for the evolution of the Cordillera Real. Pratt et al., [35] and Spikings et al., [16] proposed geological models in which new geological field observations aided into adjusted interpretations of the geological evolution of the Cordillera Real, and the model of autochthonous terrain through the mountain range. Morpho-structurally, Litherland [36] established five lithotectonic divisions consisting of belts or informal metamorphic terrains such as Guamote, Alao, Loja, Salado, and Zamora, separated by structural limits represented by the regional fault systems Peltetec, Baños Front, Llangantes fault, and Cosanga Mendez fault. Regionally, the Paute Integral hydroelectric complex is located in the Alao terrain, formed in a middle Jurassic oceanic island arc environment. It is assumed to be largely covered by extensive Plio-Pleistocene volcanic deposits, which cover much of the Ecuadorian Andes. Specifically, the hydroelectric complex is located within the Alao Paute Unit and El Pan Unit (as part of the Alao terrain). The Alao Paute Unit (which is where the Guarumales landslide lies) is made up of meta-andesites, volcanic agglomerates, tuffs, and green rocks that have developed pelite and schist facies. The structural features present regional lineaments and plans of foliation oriented to the NNE-SSW diving towards the NW [36] (see Figure A1 for a detailed geological map of the region). Locally, the Guarumales landslide is composed of two litho-stratigraphic sequences, as described later in this study. The altitude of the Guarumales landslide varies from 1300 to 1700 m.a.s.l. and the slopes range from 0° to >45°, with an average of 20°. The vegetation is considered as lower mountain rain forest, with (smaller) trees similar to those in the lowlands. Generally, the buttresses and stilt roots on trees are infrequent or non-existing [15,37]. However, it is observed that, in some locations of the Guarumales slope, anthropogenic disturbances have changed the land cover and land use (e.g., the construction of the electrical corporation of Ecuador (CELEC)'s headquarters, irrigation, and cattle grazing on the south of the slope). Other locations in the Ecuadorian Andes show similar anthropogenic disturbances (e.g., [21,38,39]). The mean annual rainfall is around 3000 mm/year measured at the Guarumales station from 2013 to 2018, with a standard deviation of ~200 mm/y. The Amazonian regime influences rainfall. The wettest season is from April to August, with 52% of the rainfall occurring during five months, and the somewhat drier season is between September and March, with 48% of the rainfall spread over seven months.

The Guarumales landslide endangers one of the most important hydropower plants of Ecuador: the Molino hydropower plant of the Paute Integral hydroelectric complex with an installed capacity of 1100 Mw. The administrative facilities of Molino are located on the Guarumales landslide, where an average of 250 people live. Most of them are employees of CELEC EP (Electrical corporation of Ecuador public company). Instrumentation and essential infrastructure are located on the landslide (Figure 1).

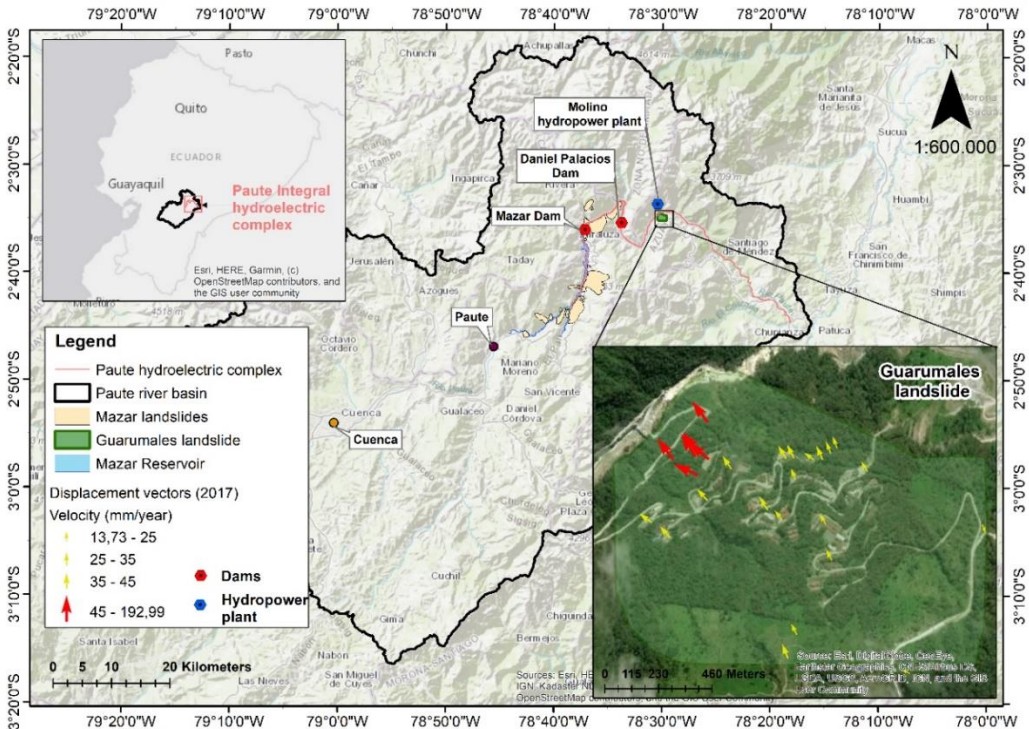

**Figure 1.** Location of the 21 landslides around Mazar and Molino hydropower facilities, including the Guarumales landslide.

## 3. Methodology and Data Availability

For this study, three geological cross sections were made in Guarumales: A-A′, B-B′, and C′-C. These were constructed from data collected during the drilling campaigns carried out from 2016–2018 and corroborated by historical data before 2016. Two of the stratigraphic columns used to construct these profiles can be seen in Figure A2. A stratigraphic correlation was used to construct the local stratigraphic sections.

Data for the surface displacement were from 2001 to 2018 and were collected monthly using total stations with 6 and 5 s of precision from 2010 to 2015, and since 2015, respectively, from a fixed reference point located on the opposite slope (2°34′ S, 78°30′ W). As a result, the spatial coordinates (x, y, z) of 26 fixed points located on the Guarumales landslide were obtained and labeled as indicated in Table 1. The cumulative horizontal displacement per observation point was determined to evaluate the slope displacement. We determined the accuracy of the measurements by comparing the residuals (using the total least squares method [40]) between the geodetical measurements and the azimuth of movement (for horizontal displacement), to the annual horizontal displacement.

A meteorological station was located south of the landslide (2°34′ S, 78°29′ W), where rainfall and evaporation data were collected daily from 2013 to 2018. Missing rainfall data were substituted by using the climatological mean of the day of the missing data, with an error of 4 mm per day [41], while the evaporation was calculated using the Penman–Monteith equation directly by the sensor using temperature, solar radiation, among other variables.

Groundwater level data were collected manually at 11 piezometers twice per month for the period of 2013 to 2018. For this study, 11 piezometers were evaluated (see Table 1 for the labels and depth information).

**Table 1.** Instrumentation in the Guarumales landslide used in this study.

| Monitoring Points | Quantity | Labels |
|---|---|---|
| Surface displacement | 26 | T4, T18, PEG3, PI3, PI5, PI6, PI9, PI10, PI11, PI12, S1, S2, S3, T9, T10, T11, T12, T13, T14, T8, T16, PI7, T17, T19, T20, and PI2. |
| Piezometric levels (with the depth of the borehole) | 11 | PP4A (43 m deep), PP4B (43 m deep), PEG3 (47 m deep), PI11 [1] (44 m deep), PP2A (77.5 m deep), PP2B (77.5 m deep), PI4 [1] (30 m deep), PI3 [1] (41 m deep), PI10 [1] (40 m deep), PI2 [1] (50 m deep), and PP3A (~50 m deep). |
| Horizontal drains | 29 | Group 1 (5 drains: 1.1, 1.2, 1.3, 1.4, and 1.7), Group 2 (15 drains: 2.0, 2.1, 2.2, 2.3, 2.4, 2.5, 2.6, 2.10, 2.11, 2.12, 2.14, 2.15, 2.16, 2.17, and 2.18), Group 4 (6 drains: 4.1, 4.2, 4.3, 4.4, 4.5, and 4.6), and Group 6 (3 drains: 6.1, 6.2, and 6.3). |
| Horizontal drains depth | | 1.1: 47 m, 1.2: 46 m, 1.3: 53 m, 1.4: 30 m, 1.7: 50 m, 2.0, 2.1: 23 m, 2.2: 29 m, 2.3: 37 m, 2.4: 25 m, 2.5: 21 m, 2.6: 18 m, 2.10: 33 m, 2.11: 12 m, 2.12: 21 m, 2.14: 34 m, 2.15: 27 m, 2.16: 43 m, 2.17: 34 m, 2.18, 4.1: 20 m, 4.2: 19 m, 4.3: 13 m, 4.4: 18 m, 4.5: 16 m, 4.6: 34 m, 6.1: 26 m, 6.2: 28 m, 6.3: 27 m. |
| Electrical conductivity | 18 | From surface water bodies (10 locations: CA-1, CA-3, CA-4, CA-5, CA-11, CA-12, CA-14, CA-15, CA-16, and CA-17) and from drains (8 locations: 1.4, 2.0, 2.5, 2.17, 2.18, 4.2, 4.5, and 6.1) |

[1] Instruments measuring piezometric levels, originally conceived as inclinometer boreholes, but adapted to measure groundwater levels.

Horizontal drains were drilled and installed in the landslide in 1994, in order to reduce the groundwater recharge in the slope [42]. In this study, the analyses were carried out using data from 2013 to 2018. Out of these, 29 were operational and formed four groups over the landslide, as indicated in Table 1. The drain discharges were measured daily, weekly, and monthly using a graduated container. Their length varied from ~30 m to 150 m.

We analyzed how the groundwater levels and drain discharges were affected by rainfall and reference evaporation [43], using transfer function noise (TFN) modeling as implemented in the Python package Pastas [44]. In general, TFN modeling was used to identify the different stresses (or forcings) that cause the input time series (e.g., head values) to fluctuate or respond. Impulse response functions were estimated to explain a time series (piezometric level and discharge from drains) based on one or more forcings (rainfall and evaporation) [44]. Outliers in the groundwater data were identified and removed using the approach of Peterson et al., [45] before the analysis.

The electrical conductivity was measured at 18 points once per week since 2018, using a multi-parameter water quality meter (Horiba U-50 series). We sampled 10 points from surface water bodies and 8 representative points from drains (1–3 per group of drains), as indicated in Table 1. Boxplot diagrams were used to assess the electrical conductivity. These were compared to each other and to the typical values found in the literature, to determine likely water sources.

The Brönnimann classification system [6] was used to provide a conceptual assessment of the possible hydrogeological mechanisms influencing mass movement. This system uses the permeability contrast between slope layers (high and low permeable), and the degree of saturation of the layers (unsaturated, saturated, and confined). Figure A9 in Appendix A.6 shows all of the possible combinations of these two parameters. The interpretations that come from analyzing the rainfall, groundwater variations, and drain discharge were linked to the geology and the spatial distribution of the measured surface displacement of the site.

An overview of the available instrumentation and locations is shown in Figure 2.

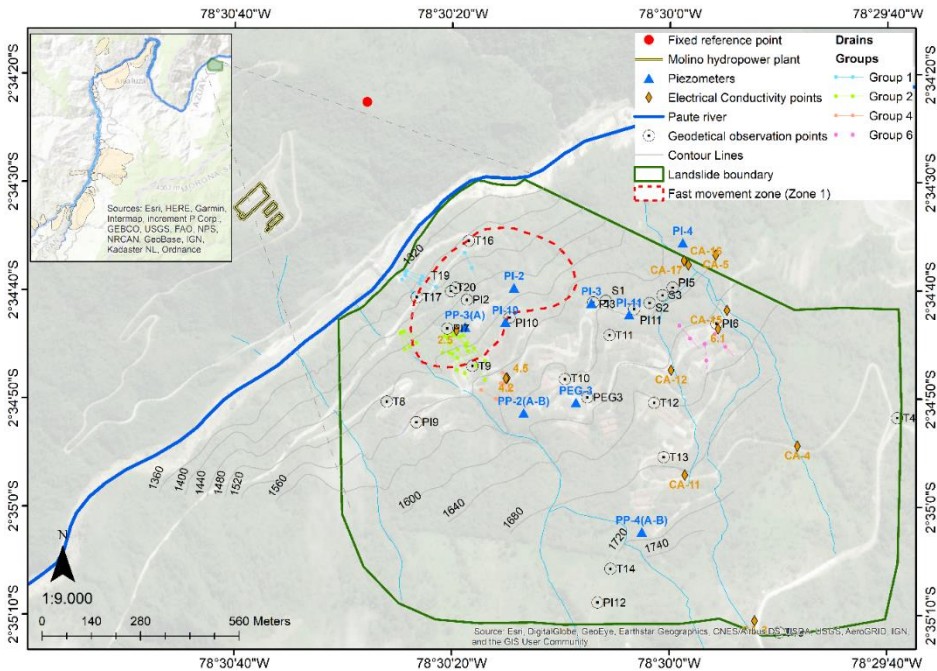

**Figure 2.** Instruments available in Guarumales and their location.

## 4. Results

### 4.1. Geology

The Guarumales landslide contained two main litho-stratigraphic units:

Unit 1—Bedrock: Undifferentiated metamorphic rock of Paleozoic–Mesozoic age, with intercalations of sericitic, chloritic, graphitic, and metavolcanic schists (shale). This lies below a slip surface composed of grey clay, containing angular clasts to sub-angles of chloritic, sericitic and graphitic schists.

Unit 2—Colluvium: Overlying the slip surface, a thick (20 to 100 m) heterogeneous colluvium layer, consisting mainly of large chloritic, sericitic and graphitic shale clasts in a clayey silt to silty sand matrix. Furthermore, in the close vicinity of the Paute river, unsorted (sub-) rounded alluvial deposits were present (see Figure 3).

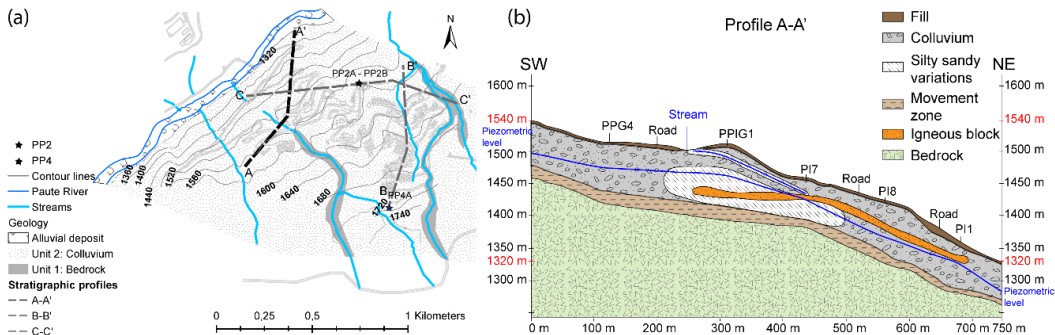

**Figure 3.** (**a**) Geological map of the Guarumales landslide, where two litho-stratigraphic units were identified. Unit 1: Bedrock, Unit 2: Colluvium, (**b**) Profile A-A' indicating the presence of the bedrock, clayey slip surface (that is located ~100 m deep), and a colluvium layer formed by shale blocks and silty sandy lenses. An igneous block was also found through the colluvium layer. Groundwater was present at two levels; one shallow presence that mainly originated in the sandy silt lens and another deeper (30 to 50 m deep) in the colluvium body. Profiles B-B' and C'-C are presented in Figures A3 and A4.

Several drilling campaigns were carried out from 2016 to 2018, and these were corroborated by data from before 2016. In addition to a deep, continuous groundwater level, a shallow groundwater level was found during the 2016–2018 drilling campaign in boreholes PP2 and PP4, suggesting that a perched groundwater level existed in a lens of sandy silt matrix, which seemed to link to small surface water streams present on the Guarumales slope (see Figure 3). Within the colluvium layer of profile A-A′, a large igneous block was found, possibly associated with the granodioritic body of Amaluza (pluton of the Eocene age) [36]. This unit is present along the Paute river, in the lower part of the basin.

### 4.2. Surface Dynamics

Survey campaigns in Guarumales take place once per month, but there are interruptions due to weather conditions resulting in an average of eight measurements per year with a minimum of three measurements per year at the 26 geodetical observation points. There are two zones with distinct movement. Zone 1 (red dashed line in Figure 2) is located in the lower part of the landslide (N-W) and represents the fastest movement with horizontal velocities that can go up to 210 mm/year. Zone 2 (yellow) represents the rest of the slope where the horizontal velocities typically range from 30 to 60 mm/year and up to 150 mm/year (Figure 4a). The cumulative yearly horizontal displacement is plotted in Figure 4b.

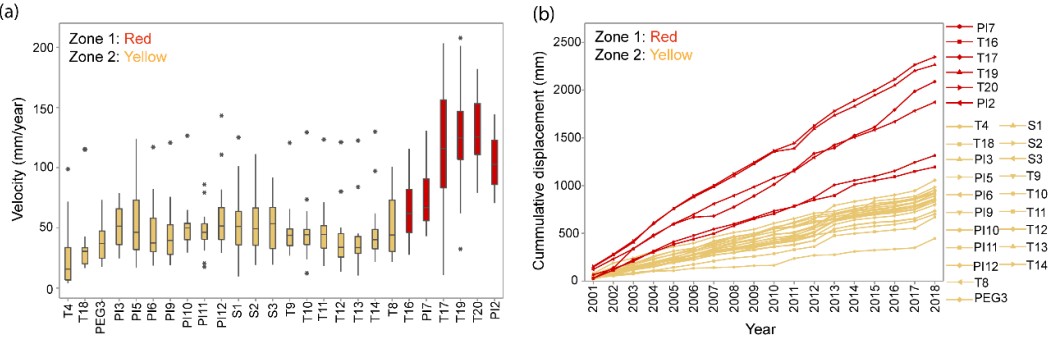

**Figure 4.** The horizontal displacement velocity of 26 geodetical observation points in Guarumales for 18 years, indicating Zone 1 (red, higher velocity) and Zone 2 (yellow, lower velocity. (**a**) Boxplots, (**b**) Cumulative horizontal displacement.

The overall direction of the movement in all 26 points was north-west, with an azimuth that varied from 250 to 330 degrees (see Figure 1). The direction and magnitude of the velocity vectors of all individual observation points along the horizontal plane are presented in Appendix A.3, in Tables A1–A4.

The annual horizontal velocity of all 26 observation points is presented for each year in Figure 5a. On average, the velocity was around 50 mm/year, which mostly corresponded to the behavior of zone 2. The higher displacement rates (expressed here as outliers) corresponded to observation points in zone 1 with velocities up to 200 mm/year in some years. In 2013, the velocities ranged from 0 to 150 mm/year with no anomalies that corresponded to Zone 1. In Figure 5b, a plan view of the evolution of the horizontal movement over 18 years is shown for PI3. It is clear that in 2013, there was a dramatic shift in the displacement; however, the effect vanished after one or two years, as the movement returned to a north-west direction.

The residuals between the observed locations and the trendline (i.e., the azimuth of movement for the horizontal coordinates), as in Figure 5b for PI3, were calculated. These results were compared to the average horizontal displacement per year to obtain the normalized residuals. This revealed that the residuals for zones 1 and 2 were ~1.7 and ~3.5 (up to six) times the average yearly displacement, respectively. To determine a reliable displacement and velocity, it was necessary to average at least 2–3 years of data to overcome the residuals. This can be seen in Appendix A.4, in Figures A5 and A6.

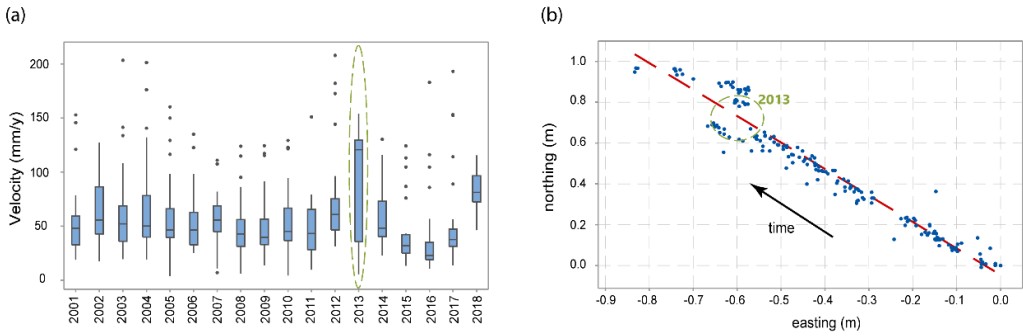

**Figure 5.** (**a**) Boxplots of the average annual horizontal displacement velocity of 26 geodetical observation points for 18 years. (**b**) Plan view of the evolution of the horizontal movement from 2001 to 2018, with the respective trendline for observation point PI3.

### 4.3. Hydro-Meteorological Analysis

The hydro-meteorological data are summarized in Figure 6. The mean annual rainfall was around 2937 mm per year. The years 2015 and 2013 were the wettest and the driest years with 3144 mm and 2607 mm of rainfall, respectively. The mean monthly rainfall ranged from 237 mm/month to 470 mm/month (Figure 6a). The wet season occurred from March to July. During the dry season, the lowest recorded monthly rainfall was 25 mm/month. The rainfall and discharge from the drains increased during the wet season (Figure 6a,c), while the evaporation increased during the drier season (Figure 6b). A time lag was always observed between the groundwater level rise (here for PP4A, depth: 43 m) and the rainfall (Figure 6d).

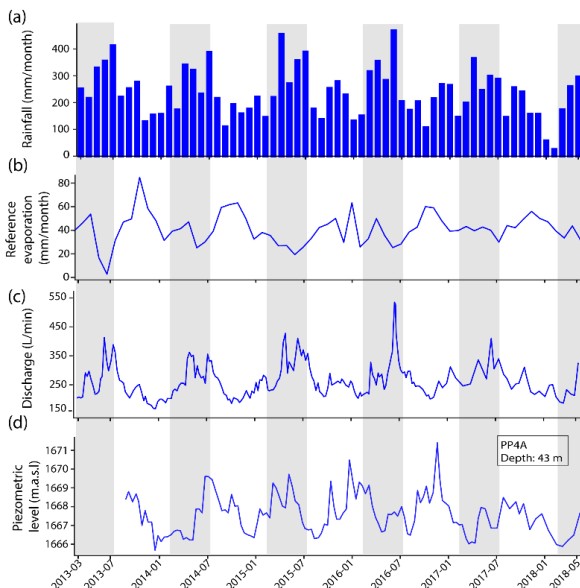

**Figure 6.** Time series of the (**a**) monthly rainfall, (**b**) monthly reference evaporation, (**c**) total discharge from 29 drains, and (**d**) piezometric level from the piezometer PP4A for the period 2013–2018. The shaded area indicates the rainy season. Time series plots for the 10 remaining piezometers and the individual 29 drains are shown in Appendix A.5, in Figures A7 and A8.

We analyzed the response of the groundwater levels to the rainfall and evaporation. We applied transfer function noise modeling with response functions using the Python package Pastas [43,44]. First, we attempted to analyze the measured groundwater levels using the rainfall and reference evaporation as stresses (forcings). The results of the analysis showed that the evaporation did not have

a significant effect on the groundwater levels, and the estimated parameters had high uncertainty. Next, the groundwater levels were analyzed with rainfall as the only stress causing groundwater fluctuations. The results were almost identical in terms of Pearson's $r^2$, and the parameters were estimated with much less uncertainty.

All 11 piezometers were analyzed, and three of them (PP2A, PP3A, and PP4A) showed a Pearson's $r^2$ higher than 0.65, indicating that measured groundwater levels in these piezometers could be analyzed reasonably well using rainfall as the only stress. The resulting response functions for these three piezometers are shown in Figure 7.

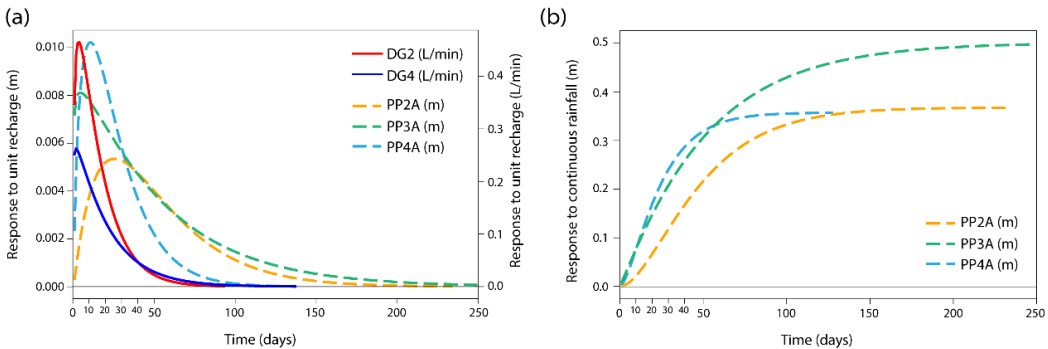

**Figure 7.** The impulse responses (**a**) for piezometers PP2A, PP3A, and PP4A, and drain groups 2 (DG2) and 4 (DG4); and the step responses (**b**) for piezometers PP2A, PP3A, and PP4A.

The impulse response represents the response of the groundwater level to an instantaneous recharge event of 1 mm/day (Figure 7a, dashed lines). The response time is defined here as the time when the impulse response reaches a peak, which represents the time lag between the rainfall event and the maximum groundwater response. The optimal value of the response time was estimated together with a 95% confidence interval. The peak response in PP2A was ~0.005 m, and the response time (time lag of the peak) was ~16–31 days. The peak response in PP3A was 0.008 m after ~1–12 days, and the peak response in PP4A was ~0.01 m after ~5–13 days.

The step response represents the level to which the head rises in response to a continuous recharge of 1 mm/day (Figure 7b). The time it takes for the response to reach its plateau is referred to as the memory of the system. Inversely, the memory of the system represents the time it takes for the effect of an impulse of rain to vanish. The optimal value of the memory was estimated together with a 95% confidence interval. In PP2A, the step response leveled off at ~0.35 m, and the memory was ~71–147 days. In PP3A, the step response leveled off at ~0.5 m after ~59–221 days, and in PP4A, the step response leveled off at ~0.35 m after ~41–76 days.

The same procedure was applied to the 29 drains located on the slope, and we attempted to analyze the measured discharge using time series analysis and rainfall as the stress. Discharge from individual drains was difficult to analyze, likely due to the relatively low discharge rates of the individual drains. Next, we attempted to analyze the cumulative discharge of each group of drains (groups 1, 2, 4, and 6 in Figure 2). The analysis for groups 2 and 4 resulted in a Pearson's $r^2$ value larger than 0.65, indicating that cumulative drain discharge could be analyzed reasonably well using rainfall as the only stress. The impulse response functions for the drain discharge are shown in Figure 7a (continuous lines). Not surprisingly, the drain response was faster than the groundwater response. A response of 0.5 L/min and a response time of ~2–5 days was obtained for Group 2 (DG2), while a response of 0.25 L/min and a response time of ~1–4 days was obtained for Group 4 (DG4).

The electrical conductivity was tested in 18 locations across the slope and is shown in Figure 8. Values from the samples collected from the surface water bodies and drains were similar, ranging from ~20 μs/cm to ~90–100 μs/cm (see Figure 2). These values are typical for rainfall water. The variation of the electrical conductivity values was larger in the surface water bodies than in the drains.

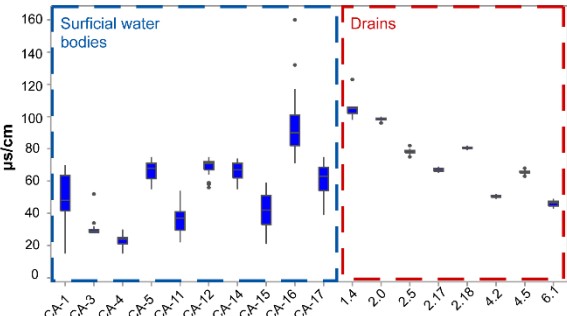

**Figure 8.** The electrical conductivity for 18 observation points in the slope for the period July–December/2018.

## 5. A Hydrogeological Conceptual Model of Guarumales Slope

Guarumales is a deep-seated landslide in a local complex geological setting. Here, we use the Brönnimann classification system to interpret the hydrogeological characteristics of the slow-moving landslide (see Appendix A.6). Based on all observations, we simplified the landslide in a two-layer slope composed of a relatively permeable and mainly unsaturated colluvium layer and a low permeable, saturated bedrock layer with a 1–3 m thick weak layer acting as slip surface between the colluvium and the bedrock. Furthermore, we observed a delayed but distinct correlation between the rainwater input and groundwater response of the unconfined aquifer in the slope. The heterogeneous nature of the colluvium and bedrock layers, highly fractured and weathered in some locations while intact in others, added complexity, including local perched water tables. Lastly, we found that the drains in the slope did not tap into the deeper groundwater system but drained the perched areas of the otherwise unsaturated top layer. This places the Guarumales slope in hydrogeological class B1 (see Appendix A.6), mainly influenced by local infiltration and percolation processes, free-draining into the Paute river at the toe of the slope, and with limited influence from the deep regional groundwater flow (Figure 9).

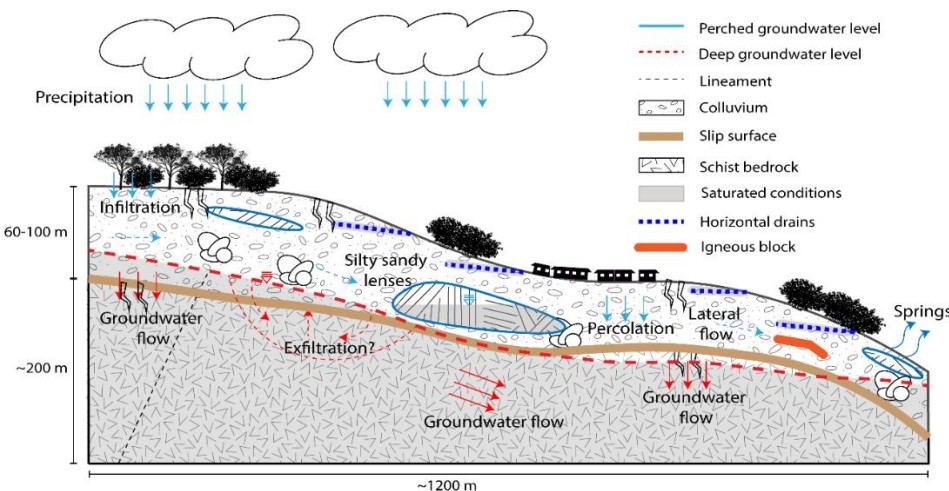

**Figure 9.** Conceptual model of the Guarumales landslide. There is a deep permanent groundwater level that varies around the slip surface, with scattered perched groundwater systems in silty sandy lenses underlaid by low permeable blocks. The Guarumales slope is an unconfined system, which means there is no pressurized water and very limited upward flow of water. There are 29 horizontal drains, which do not reach the deep groundwater systems but drain small perched water bodies in the slope, which are fed by rainfall, as also evidenced by the very low electrical conductivity of the drained water.

In terms of comparing the displacement to hydro-meteorological factors, we concluded from the data shown in Figure 5b that no seasonal signal in the displacement could be determined due to the coarse time resolution and measurement error, which means that no relation could be established between the rainfall and displacement at this stage.

## 6. Discussion

In this study, we analyzed the role of predisposing conditions and triggering factors on the stability of the Guarumales landslide. Eighteen years of displacement measurements showed that the yearly surface displacement rates were constant over time. While rainfall was shown to strongly influence the unconfined groundwater system with a ~1–31 days response time, no monthly variations in the horizontal displacement rates were observed with our displacement data. There were some areas where secondary shallow landslides and debris flow happened on top of the deep-seated, slow-moving landslide, and these were intensified during high-intensity rainfall, blocking small streams and roads in Guarumales [32]. These movements occurred on a short time scale and could not be detected by the current monitoring system. Therefore, it was not possible to link this behavior to the proposed triggering factors.

The slow-moving behavior of the Guarumales landslide and its spatial distribution showed a constant annual deformation rate over 18 years, based on (bi-) monthly surveys. This was also reported by others [46–51]. However, the limited subsurface information did not allow for detailed geological or geotechnical interpretation of the spatially distributed character of the Guarumales landslide. The deformation patterns did not link with the measured groundwater levels. The different compositions of the lithology and the depths of the slip surface (30 to >90 m) may influence the movement patterns of the slope. Therefore, geophysical monitoring in Guarumales could be a useful tool to identify subsurface structures.

Two additional limitations were identified with the 18 years of geodetical data. First, the limited temporal resolution of, on average, eight measurements per year, was insufficient to identify small accelerations and decelerations of surface displacement following the groundwater response to rainfall, if present. This is especially important as our analyses showed that groundwater fluctuations were correlated with rainfall with a response time of the maximum head after rainfall of ~1 to 31 days, and an increased head that lasted ~40 to 220 days. Second, the geodetical accuracy (1–5 cm) was low compared to the measured displacement (1 to 10 cm/year), which complicated the interpretation of the results. For most of the landslide, multi-annual displacement was required to assess the displacement magnitude and direction, which made it impossible to link the displacement to potential driving processes such as specific rainfall periods. To unravel the transient behavior of the Guarumales landslide, we need higher frequency and more accurate displacement measurements, which may be obtained with real-time kinematic global positioning systems (RTK-dGNSS).

In 2013, a shift in the direction of movement was found in all surface monitoring points in Guarumales. What influenced this directional shift is not clear. The cause could be related to a displacement of the fixed reference point on the other side of the valley from where the geodetical measurements were taken. Alternatively, it could be related to high seismic activity in the study area where the number of seismic events (115), with a magnitude above 4 Mb (body-wave magnitude), was higher in 2013 than in other years [52]. The additional lateral forces would have weakened the soil by reducing its shear strength. Additionally, an alteration of the drainage system at the toe of the landslide through the reinforcement of an existing retaining wall was conducted in 2013, which may have resulted in a temporary direction shift of the overall movement. The excess of water that built up at the toe of the slope was unable to drain properly, and this groundwater accumulation, which was evidenced in the springs found at the toe of the slope, decreased the stability of the slope. After the modification of the retaining walls, water may have found a new way to drain towards the river, resulting in a normalization of the movement to the north-west direction.

The 2016–2018 drillings revealed that the slip surface of the Guarumales landslide was situated between 30 m and >90 m below the ground surface. This is deeper than was reported previously [32,53,54] based on the analysis of inclinometer data from eight inclinometers located in the landslide, in the periods of 1994–1996, 1994–1998, and 2000–2001, respectively. The recent explorations revealed that large blocks (~20 m in diameter), abundantly present in the colluvium layer, were mistakenly considered as bedrock. This misunderstanding led to relatively shallow drillings, and inclinometers were installed with their lowest point fixed in the moving block instead of in the stable bedrock, which compromised the inclinometer results.

The observed groundwater level fluctuations of ~0.6 m to ~4 m at Guarumales were small compared to the depth of the slip surface (~30 m to >90 m). With a groundwater depth of 30 m, a 1 m groundwater level rise equals a relative increase in pore pressure of 1%, which has a limited effect on the overall slope movement. Hydrologically, a 1-m groundwater level rise is equivalent to 10–100 mm of groundwater recharge (assuming an effective porosity of 1–10%). This was the order of magnitude of groundwater recharge in deep-seated landslides observed by others [9,55,56]. This behavior was observed during the 6 years of groundwater monitoring, in which time, the landslide movement did not change its annual displacement rate.

We analyzed groundwater heads at three observation points, with rainfall as a driving force. The response time for PP3A and PP4A was shorter than for PP2A, while the memory of the system for all three wells was similar, where PP4A had a slightly shorter memory. It was not possible to analyze the measured heads with a time series analysis at all observation points. One reason may be that the applied approach was linear, which may be insufficient for deep groundwater levels, where non-linear effects may be important [44].

Groundwater observations could not be analyzed with time series analysis in eight of the eleven piezometers. Time series analysis was carried out using rainfall as the only driving force and approximating the system as linear (i.e., twice the rainfall results in twice the rise in head). This approach was not adequate when the response to rainfall was non-linear in, for example, fractured and fissured rock formations. Some of the measured heads showed unexpected jumps, which could not be explained physically and were likely related to instrument handling issues (PEG3, PP2B). Other measured heads showed only very small variations over the entire measurement period (PI10, PP4B), which may indicate the possible presence of clogged screens [57] or that the piezometer was screened in a layer of very low permeability. These responses come from boreholes that were meant to host inclinometers, but they were adapted to provide piezometric level measurements. As the installation was not optimal, this could have resulted in limited connection to the groundwater body (perched or deep), and the readings will have a larger uncertainty (due to the inconsistencies in the installation of pipes or screens within the borehole). Instead, they were considered as indicative for possible follow-up installation of piezometers, not for detailed groundwater monitoring.

An interesting aspect is the limited functioning of the 29 horizontal drains. In 1998, the drains produced only 20% of the water initially discharged in 1994 [32]; this is ~12% of the annual rainfall, compared to an initial yield of ~56% of the annual rainfall. Since the groundwater heads started to be collected sometime after the installation of the drains, the effect of drainage could not be detected in the groundwater observations. Nevertheless, time series analysis revealed that if the drains were not 12% of the total annual rainfall, the groundwater recharge would increase by ~1 mm/day. The additional recharge would cause a rise in the groundwater level of up to ~0.5 m in the wells PP2A, PP3A, and PP4A. This was concluded from Figure 7b, where the step response due to a constant recharge of 1 mm/day resulted in a groundwater level rise of ~0.3 to ~0.5 m. The drains did not reach the deep groundwater systems but drained small perched water bodies within the soil fed by rainfall. This was confirmed by the electrical conductivity results at the drains, which were similar to electrical conductivity values of rainfall.

The Brönnimann classification system allowed us to simplify the Guarumales landslide to a two-layer system, separated by a slip surface (~30 to >90 m deep). This helped with interpreting

the conceptual model where the bottom layer (schist bedrock) was permanently saturated, and the colluvium layer was unsaturated. The permeability of the slope was heterogeneous. Highly weathered material and relatively intact material were found in the slope during the drilling campaigns in both the bedrock and colluvium. We found perched groundwater in the slope, as well as deep/permanent groundwater. Infiltration from rainfall and a regional groundwater system were their main sources.

## 7. Conclusions

The objective of this study was to characterize, understand, and quantify the mechanisms interfering with the stability of the Guarumales landslide. Special attention was paid to the role of predisposing conditions and possible triggering factors, such as rainfall and groundwater fluctuations. We showed that the movement of the landslide was continuous on an annual timescale, both in the direction and horizontal displacement rate, and there were no significant changes over the last 18 years. Faster movement was found at the toe of the slope where clayey silt lenses, silty sand lenses, and springs were common. A detailed response of the displacement rate to pore pressure changes, if any, could not be determined as the surface displacement records were not detailed enough.

A conceptual model was developed for the Guarumales landslide. Shallow perched groundwater levels were located in clayey silt and silty sand lenses, which were part of the permeable and mainly unsaturated colluvium layer. The colluvium layer contained blocks of highly fractured shales, overlying a largely saturated unconfined bedrock layer. This agreed with the heterogeneous nature of the slope, which included both highly fractured material and intact material. The unconfined groundwater system is responded with a ~1 to 31 day response time to rainfall forcing on the piezometers PP3A, PP4A, and PP2A, at depths of ~30, ~40, and ~47 m, respectively. The pore water fluctuations were too small to have a significant effect on the landslide movement. The existing drainage did not reach the deep groundwater system. The system only drained small perched water bodies fed by rainfall, as evidenced by the quick response of drain discharge to rainfall, and the low values of electrical conductivity of the drained water.

Future work should be aimed at investigating whether pore water fluctuations in the Guarumales landslide have a significant effect on fluctuations in the landslide movement. This would require the collection of displacement and groundwater level data with a higher spatio-temporal accuracy and resolution by using remote sensing or real-time kinematic global positioning systems (RTK-dGNSS) acquisitions.

**Author Contributions:** Conceptualization, A.U.V., J.R., P.G. and T.B.; Data curation, A.U.V. and J.R.; Formal analysis, A.U.V. and J.R.; Investigation, J.R.; Methodology, A.U.V., J.R. and M.B.; Software, M.B.; Supervision, M.B., P.G. and T.B.; Validation, M.B., P.G. and T.B.; Visualization, A.U.V. and J.R.; Writing—original draft, A.U.V.; Writing—review & editing, M.B., P.G. and T.B. All authors have read and agreed to the published version of the manuscript.

**Funding:** This work was funded by Secretaría de Educación Superior, Ciencia, Tecnología e Innovación, SENESCYT Ecuador, grant "Convocatoria Abierta 2017–Componente General".

**Acknowledgments:** We thank Corporacion Electrica del Ecuador CELEC EP and Hidropaute business unit team for the collaboration during the development of this project and for providing the database that was used for the analysis and interpretations. A special mention to R. Guerrero, who processed the rainfall data that was finally used in this analysis. The authors would also like to acknowledge the contribution of J. Uwihirwe for her insightful discussions.

**Conflicts of Interest:** The authors declare no conflict of interest.

# Appendix A

*Appendix A.1. Geological Map of the Paute Integral Hydroelectric Complex*

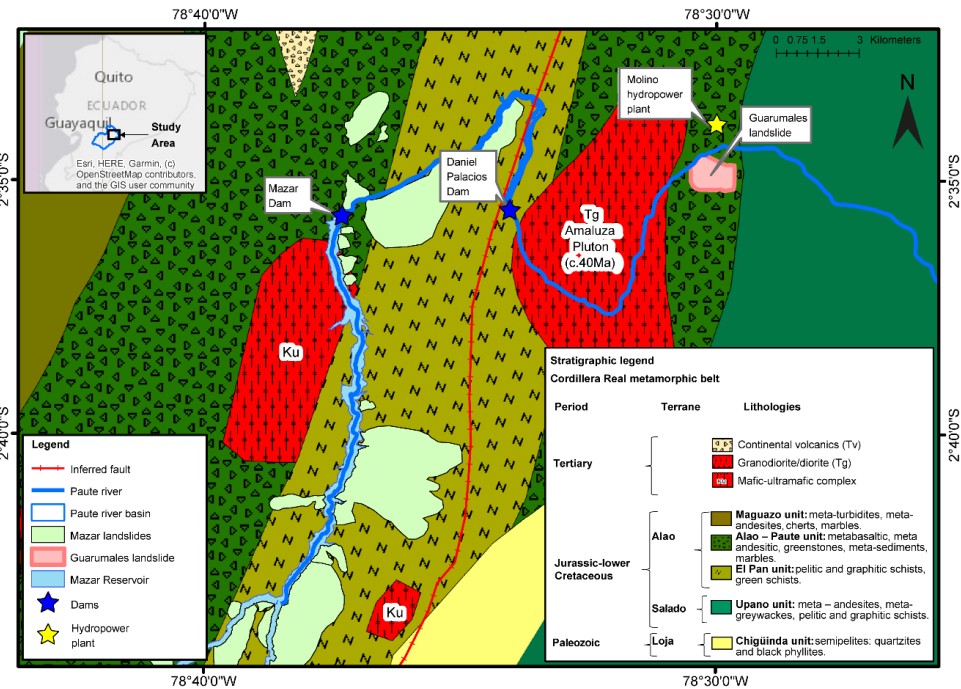

**Figure A1.** Geological map of the Paute Integral hydroelectric complex.

*Appendix A.2. The Stratigraphic Columns PPG-1 and PPG-2, and the Stratigraphic Profiles B-B′ and C′-C*

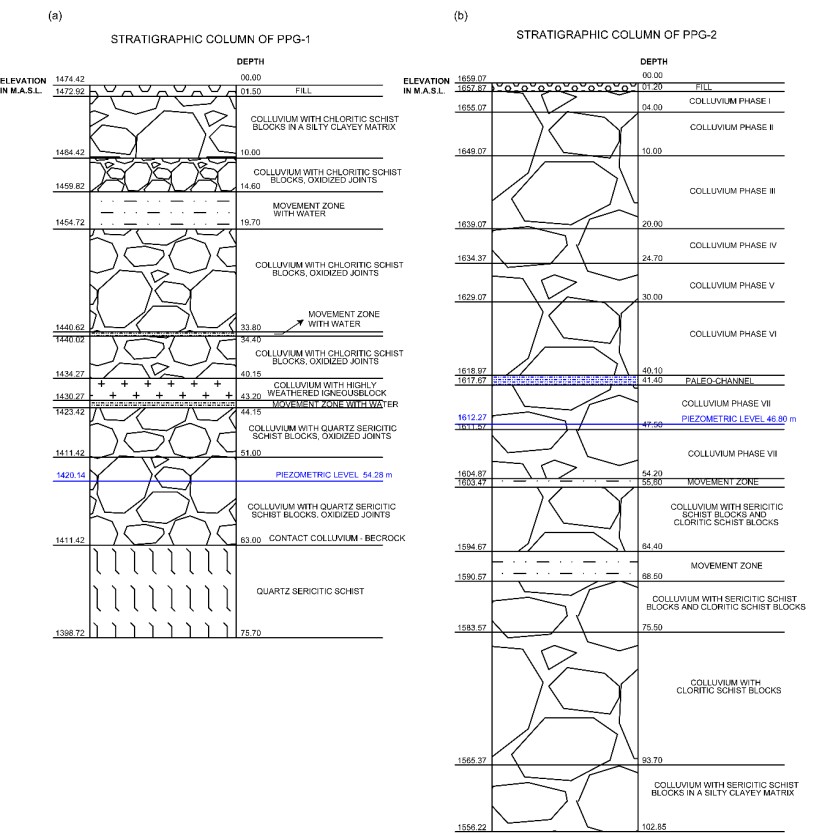

**Figure A2.** Stratigraphic columns for boreholes PPG-1 (**a**) and PPG-2 (**b**).

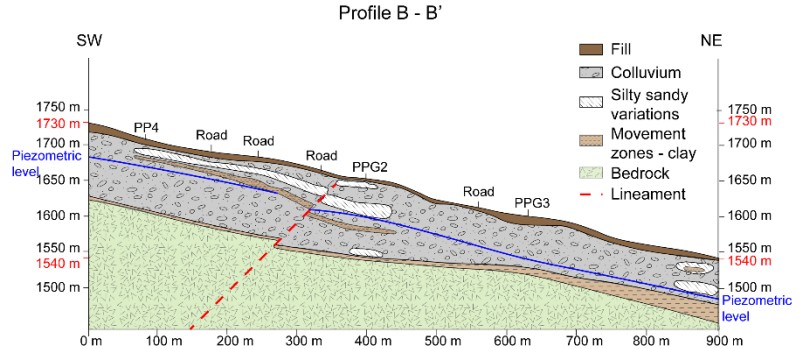

**Figure A3.** Stratigraphic profile B-B′.

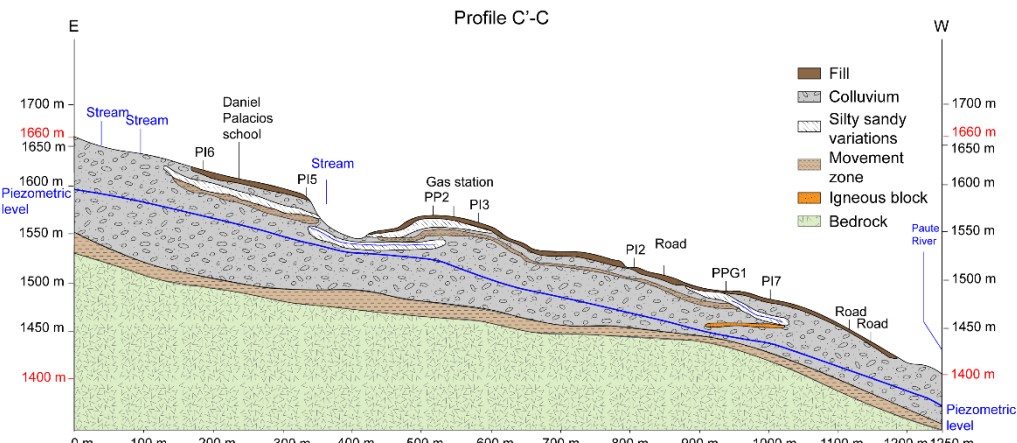

**Figure A4.** Stratigraphic profile C′-C.

*Appendix A.3. Displacement and Azimuth of Movement of 26 Fixed Points in the Guarumales Landslide from 2001 to 2018*

**Table A1.** The initial coordinates and horizontal displacement detected by 26 fixed points in Guarumales from 2001 to 2013.

| Point | Initial Coordinates (m) | | | Displacement (mm) | | | | | | | | | | | | |
|---|---|---|---|---|---|---|---|---|---|---|---|---|---|---|---|---|
| | x | y | z | 2001 | 2002 | 2003 | 2004 | 2005 | 2006 | 2007 | 2008 | 2009 | 2010 | 2011 | 2012 | 2013 |
| PEG3 | 778008.63 | 9714864.70 | 1590.91 | 46.19 | 55.15 | 20.46 | 58.42 | 37.61 | 31.44 | 24.83 | 23.75 | 50.94 | 41.19 | 43.82 | 46.13 | 36.33 |
| PI3 | 778025.19 | 9715134.72 | 1556.52 | 66.21 | 78.41 | 36.59 | 78.81 | 52.80 | 32.69 | 50.21 | 62.42 | 41.65 | 52.87 | 63.87 | 66.25 | 24.60 |
| PI5 | 778251.24 | 9715176.83 | 1540.81 | 18.98 | 83.36 | 54.38 | 42.23 | 40.59 | 78.65 | 71.21 | 46.30 | 45.62 | 66.10 | 16.76 | 46.52 | 33.11 |
| PI6 | 778376.07 | 9715074.17 | 1609.79 | 50.80 | 56.79 | 25.84 | 49.93 | 33.47 | 61.88 | 18.71 | 39.47 | 37.11 | 37.39 | 22.95 | 73.40 | 117.09 |
| PI9 | 777522.50 | 9714794.21 | 1531.74 | 33.85 | 37.18 | 53.14 | 30.50 | 42.20 | 43.37 | 57.99 | 38.66 | 25.86 | 25.86 | 28.67 | 52.41 | 120.74 |
| PI10 | 777787.84 | 9715091.17 | 1501.97 | 49.72 | 52.02 | 52.86 | 45.24 | 50.25 | 55.44 | 53.20 | 42.21 | 40.17 | 41.25 | 29.56 | 64.85 | 126.58 |
| PI11 | 778141.91 | 9715116.04 | 1551.38 | 50.35 | 86.01 | 43.39 | 53.42 | 43.73 | 44.53 | 52.48 | 44.00 | 53.59 | 43.97 | 59.24 | 52.61 | 17.65 |
| PI12 | 778037.53 | 9714282.10 | 1786.93 | 52.22 | 56.10 | 65.48 | 50.98 | 58.19 | 53.08 | 81.91 | 46.83 | 32.98 | 49.32 | 29.23 | 70.67 | 143.22 |
| S1 | 778061.23 | 9715138.95 | 1556.09 | 20.57 | 47.61 | 55.35 | 50.36 | 51.98 | 45.72 | 68.34 | 52.96 | 36.97 | 63.29 | 9.70 | 65.14 | 125.24 |
| S2 | 778185.72 | 9715133.53 | 1550.76 | 78.28 | 111.23 | 49.84 | 73.11 | 47.32 | 37.36 | 64.40 | 46.51 | 32.76 | 63.68 | 49.50 | 55.00 | 21.33 |
| S3 | 778222.45 | 9715155.27 | 1546.36 | 57.09 | 69.85 | 58.53 | 66.55 | 60.79 | 30.13 | 69.94 | 43.11 | 39.27 | 68.12 | 43.35 | 56.64 | 24.36 |
| T9 | 777682.11 | 9714953.91 | 1512.26 | 46.35 | 40.54 | 51.25 | 40.19 | 45.66 | 47.26 | 48.30 | 27.78 | 40.02 | 42.17 | 29.45 | 56.63 | 120.66 |
| T10 | 777944.87 | 9714916.40 | 1572.02 | 53.71 | 44.98 | 47.14 | 43.68 | 45.35 | 32.41 | 63.88 | 39.09 | 37.30 | 46.53 | 12.29 | 40.00 | 129.41 |
| T11 | 778071.91 | 9715041.24 | 1585.34 | 51.87 | 45.46 | 49.45 | 43.52 | 40.06 | 52.02 | 51.14 | 33.19 | 45.86 | 34.77 | 32.54 | 67.23 | 123.38 |
| T12 | 778198.74 | 9714849.28 | 1654.86 | 44.57 | 50.20 | 34.27 | 33.67 | 31.44 | 31.79 | 44.95 | 25.18 | 33.99 | 29.63 | 16.86 | 46.04 | 121.19 |
| T13 | 778225.66 | 9714694.89 | 1678.90 | 45.32 | 42.23 | 29.26 | 40.33 | 32.62 | 34.71 | 44.49 | 30.03 | 26.20 | 31.35 | 27.46 | 39.27 | 122.58 |
| T14 | 778074.50 | 9714377.62 | 1752.03 | 37.66 | 42.89 | 39.64 | 39.22 | 44.84 | 41.70 | 62.02 | 33.72 | 24.42 | 40.77 | 47.25 | 53.31 | 129.95 |
| T4 | 778889.60 | 9714805.92 | 1823.11 | 35.49 | 17.49 | 19.50 | 31.24 | 3.91 | 24.97 | 7.03 | 6.14 | 13.76 | 4.35 | 71.79 | 33.08 | 5.56 |
| T8 | 777438.58 | 9714852.99 | 1500.13 | 26.67 | 34.74 | 54.22 | 29.38 | 48.73 | 54.06 | 45.29 | 30.62 | 32.88 | 38.69 | 79.08 | 80.90 | 73.70 |
| T18 | 778553.43 | 9714196.82 | 1892.91 | 29.78 | 26.63 | 32.96 | 19.24 | 32.61 | 32.41 | 34.97 | 31.47 | 17.42 | 28.02 | 42.89 | 31.14 | 115.01 |
| PI7 | 777610.35 | 9715060.41 | 1461.22 | 66.47 | 59.77 | 78.50 | 107.92 | 98.07 | 65.65 | 67.04 | 53.86 | 64.75 | 71.49 | 45.00 | 96.26 | 130.92 |
| T16 | 777671.99 | 9715309.82 | 1380.58 | 27.67 | 87.32 | 108.30 | 78.00 | 81.95 | 55.71 | 59.17 | 82.33 | 69.80 | 55.01 | 79.38 | 64.78 | 45.54 |
| T17 | 777524.25 | 9715149.42 | 1397.36 | 28.33 | 107.17 | 203.31 | 140.60 | 116.35 | 73.09 | 10.93 | 95.05 | 115.37 | 121.47 | 150.96 | 172.51 | 57.91 |
| T19 | 777634.77 | 9715175.96 | 1436.38 | 145.69 | 125.24 | 133.64 | 201.13 | 149.93 | 122.59 | 107.65 | 114.74 | 124.29 | 129.19 | 32.46 | 207.84 | 139.49 |
| T20 | 777621.50 | 9715166.31 | 1435.48 | 152.79 | 127.29 | 141.44 | 176.08 | 160.29 | 134.89 | 107.19 | 123.84 | 117.14 | 122.82 | 79.19 | 182.05 | 153.90 |
| PI2 | 777666.36 | 9715141.99 | 1457.73 | 120.92 | 109.00 | 107.66 | 131.97 | 129.17 | 98.00 | 110.92 | 86.11 | 91.29 | 94.44 | 70.64 | 144.16 | 129.77 |

**Table A2.** The initial coordinates and horizontal displacement detected by 26 fixed points in Guarumales from 2014 to 2018.

| Point | Initial Coordinates (m) | | | Displacement (mm) | | | | | | |
|---|---|---|---|---|---|---|---|---|---|---|
| | x | y | z | 2014 | 2015 | 2016 | 2017 | 2018 | Average | Accumulated |
| PEG3 | 778008.63 | 9714864.70 | 1590.91 | 33.84 | 24.52 | 17.71 | 34.60 | 73.11 | 38.89 | 700.05 |
| PI3 | 778025.19 | 9715134.72 | 1556.52 | 47.50 | 34.83 | 24.95 | 38.44 | 65.64 | 51.04 | 918.76 |
| PI5 | 778251.24 | 9715176.83 | 1540.81 | 69.83 | 124.20 | 19.77 | 28.73 | 96.17 | 54.59 | 982.54 |
| PI6 | 778376.07 | 9715074.17 | 1609.79 | 37.59 | 31.62 | 18.76 | 33.68 | 82.10 | 46.03 | 828.58 |
| PI9 | 777522.50 | 9714794.21 | 1531.74 | 48.68 | 22.85 | 19.11 | 40.63 | 75.70 | 44.30 | 797.39 |
| PI10 | 777787.84 | 9715091.17 | 1501.97 | 52.80 | 34.52 | 33.06 | 40.28 | 62.04 | 51.45 | 926.07 |
| PI11 | 778141.91 | 9715116.04 | 1551.38 | 48.31 | 31.00 | 20.45 | 31.79 | 79.38 | 47.55 | 855.91 |
| PI12 | 778037.53 | 9714282.10 | 1786.93 | 42.41 | 34.17 | 33.65 | 44.31 | 110.84 | 58.64 | 1055.58 |
| S1 | 778061.23 | 9715138.95 | 1556.09 | 60.79 | 32.61 | 27.23 | 38.14 | 101.14 | 52.95 | 953.15 |
| S2 | 778185.72 | 9715133.53 | 1550.76 | 48.99 | 32.09 | 19.23 | 36.45 | 87.24 | 53.02 | 954.29 |
| S3 | 778222.45 | 9715155.27 | 1546.36 | 50.26 | 32.43 | 19.39 | 33.02 | 91.79 | 50.81 | 914.62 |
| T9 | 777682.11 | 9714953.91 | 1512.26 | 47.98 | 28.75 | 26.88 | 37.23 | 65.65 | 46.82 | 842.75 |
| T10 | 777944.87 | 9714916.40 | 1572.02 | 45.38 | 30.38 | 23.90 | 38.29 | 73.58 | 47.07 | 847.31 |
| T11 | 778071.91 | 9715041.24 | 1585.34 | 39.63 | 29.60 | 18.23 | 32.87 | 71.18 | 47.89 | 861.99 |
| T12 | 778198.74 | 9714849.28 | 1654.86 | 44.95 | 24.43 | 13.87 | 25.99 | 80.18 | 40.73 | 733.20 |
| T13 | 778225.66 | 9714694.89 | 1678.90 | 40.32 | 25.21 | 10.53 | 29.67 | 83.91 | 40.86 | 735.46 |
| T14 | 778074.50 | 9714377.62 | 1752.03 | 37.80 | 26.88 | 21.81 | 29.38 | 97.20 | 47.25 | 850.47 |
| T4 | 778889.60 | 9714805.92 | 1823.11 | 34.98 | 13.33 | 11.29 | 13.73 | 98.83 | 24.80 | 446.47 |
| T8 | 777438.58 | 9714852.99 | 1500.13 | 100.79 | 21.58 | 24.09 | 42.95 | 72.81 | 49.51 | 891.20 |
| T18 | 778553.43 | 9714196.82 | 1892.91 | 22.79 | 18.18 | 20.39 | 16.62 | 115.49 | 37.11 | 668.01 |
| PI7 | 777610.35 | 9715060.41 | 1461.22 | 46.81 | 43.16 | 56.60 | 88.91 | 73.54 | 73.04 | 1314.72 |
| T16 | 777671.99 | 9715309.82 | 1380.58 | 115.90 | 41.88 | 38.42 | 56.13 | 46.28 | 66.31 | 1193.57 |
| T17 | 777524.25 | 9715149.42 | 1397.36 | 130.17 | 86.75 | 182.82 | 193.00 | 102.26 | 116.00 | 2088.06 |
| T19 | 777634.77 | 9715175.96 | 1436.38 | 96.57 | 113.01 | 104.57 | 153.07 | 62.15 | 125.73 | 2263.23 |
| T20 | 777621.50 | 9715166.31 | 1435.48 | 112.07 | 104.59 | 116.05 | 152.27 | 83.38 | 130.40 | 2347.28 |
| PI2 | 777666.36 | 9715141.99 | 1457.73 | 82.36 | 76.05 | 85.80 | 114.09 | 90.67 | 104.06 | 1873.01 |

**Table A3.** The initial coordinates and azimuth of movement detected by 26 fixed points in Guarumales from 2001 to 2013.

| Point | Initial Coordinates (m) | | | Azimuth (Degrees) | | | | | | | | | | | | |
|---|---|---|---|---|---|---|---|---|---|---|---|---|---|---|---|---|
| | x | y | z | 2001 | 2002 | 2003 | 2004 | 2005 | 2006 | 2007 | 2008 | 2009 | 2010 | 2011 | 2012 | 2013 |
| PEG3 | 778008.63 | 9714864.70 | 1590.91 | 341.94 | 321.72 | 333.18 | 332.40 | 290.78 | 312.40 | 327.50 | 338.25 | 338.03 | 336.37 | 31.23 | 348.50 | 312.77 |
| PI3 | 778025.19 | 9715134.72 | 1556.52 | 341.91 | 327.10 | 330.34 | 341.39 | 319.66 | 351.52 | 339.75 | 337.00 | 326.89 | 332.80 | 16.26 | 352.09 | 327.00 |
| PI5 | 778251.24 | 9715176.83 | 1540.81 | 267.82 | 312.33 | 339.04 | 337.44 | 328.73 | 329.84 | 327.02 | 343.36 | 286.00 | 344.31 | 220.66 | 314.60 | 329.29 |
| PI6 | 778376.07 | 9715074.17 | 1609.79 | 307.47 | 322.07 | 350.30 | 329.22 | 309.98 | 334.79 | 307.57 | 323.90 | 339.16 | 318.69 | 200.18 | 343.62 | 33.30 |
| PI9 | 777522.50 | 9714794.21 | 1531.74 | 321.45 | 314.96 | 348.28 | 319.10 | 320.52 | 332.46 | 337.26 | 344.78 | 310.33 | 328.47 | 226.78 | 301.07 | 29.64 |
| PI10 | 777787.84 | 9715091.17 | 1501.97 | 315.11 | 323.42 | 353.07 | 317.43 | 328.58 | 337.21 | 335.18 | 342.88 | 312.81 | 333.33 | 256.21 | 316.59 | 29.96 |
| PI11 | 778141.91 | 9715116.04 | 1551.38 | 196.94 | 318.36 | 339.36 | 334.25 | 314.75 | 337.71 | 337.27 | 339.80 | 341.01 | 21.69 | 17.59 | 9.81 | 312.88 |
| PI12 | 778037.53 | 9714282.10 | 1786.93 | 337.53 | 339.81 | 345.56 | 353.32 | 326.15 | 346.48 | 339.26 | 338.17 | 335.33 | 336.49 | 210.71 | 327.00 | 25.07 |
| S1 | 778061.23 | 9715138.95 | 1556.09 | 287.32 | 325.80 | 348.16 | 325.67 | 330.05 | 343.31 | 343.07 | 343.88 | 322.19 | 325.21 | 67.14 | 317.25 | 30.02 |
| S2 | 778185.72 | 9715133.53 | 1550.76 | 176.95 | 304.27 | 355.14 | 349.45 | 317.58 | 333.10 | 333.79 | 349.58 | 296.17 | 350.55 | 14.91 | 2.74 | 324.67 |
| S3 | 778222.45 | 9715155.27 | 1546.36 | 172.17 | 292.50 | 332.17 | 347.07 | 327.98 | 310.71 | 335.42 | 350.06 | 299.67 | 348.56 | 13.02 | 9.45 | 328.77 |
| T9 | 777682.11 | 9714953.91 | 1512.26 | 301.39 | 322.39 | 350.89 | 313.61 | 316.84 | 347.38 | 335.86 | 332.63 | 320.70 | 327.45 | 220.29 | 308.28 | 30.12 |
| T10 | 777944.87 | 9714916.40 | 1572.02 | 310.38 | 316.23 | 349.52 | 319.98 | 320.51 | 358.54 | 323.79 | 330.74 | 330.46 | 329.76 | 261.32 | 287.30 | 24.87 |
| T11 | 778071.91 | 9715041.24 | 1585.34 | 316.51 | 318.21 | 356.99 | 322.66 | 325.76 | 338.86 | 342.92 | 325.47 | 338.29 | 326.18 | 230.90 | 348.87 | 31.34 |
| T12 | 778198.74 | 9714849.28 | 1654.86 | 296.45 | 327.41 | 357.37 | 333.89 | 310.05 | 330.33 | 340.37 | 330.36 | 330.04 | 325.63 | 198.11 | 323.64 | 33.00 |
| T13 | 778225.66 | 9714694.89 | 1678.90 | 319.42 | 336.11 | 351.86 | 345.72 | 299.14 | 343.58 | 335.44 | 333.85 | 331.29 | 342.21 | 198.62 | 321.75 | 31.72 |
| T14 | 778074.50 | 9714377.62 | 1752.03 | 319.20 | 351.88 | 330.65 | 346.46 | 338.46 | 336.57 | 340.41 | 339.06 | 315.66 | 344.70 | 191.31 | 325.78 | 28.87 |
| T4 | 778889.60 | 9714805.92 | 1823.11 | 26.53 | 328.51 | 198.46 | 6.06 | 284.64 | 161.34 | 234.05 | 61.74 | 5.36 | 175.66 | 28.53 | 69.90 | 152.51 |
| T8 | 777438.58 | 9714852.99 | 1500.13 | 308.93 | 305.84 | 344.99 | 310.03 | 320.04 | 340.33 | 317.57 | 341.48 | 315.59 | 322.91 | 231.42 | 269.06 | 18.12 |
| T18 | 778553.43 | 9714196.82 | 1892.91 | 359.14 | 303.30 | 314.99 | 296.83 | 315.97 | 329.82 | 300.04 | 297.98 | 313.24 | 321.44 | 193.36 | 284.86 | 31.59 |
| PI7 | 777610.35 | 9715060.41 | 1461.22 | 304.29 | 310.47 | 345.76 | 309.24 | 316.41 | 323.09 | 329.02 | 321.21 | 313.94 | 323.30 | 270.92 | 311.42 | 25.05 |
| T16 | 777671.99 | 9715309.82 | 1380.58 | 178.85 | 332.05 | 335.34 | 342.48 | 327.98 | 323.53 | 313.43 | 342.29 | 336.99 | 332.41 | 328.68 | 349.62 | 254.72 |
| T17 | 777524.25 | 9715149.42 | 1397.36 | 232.57 | 338.04 | 332.99 | 336.97 | 318.22 | 317.92 | 286.51 | 324.92 | 337.51 | 326.79 | 335.59 | 330.77 | 274.92 |
| T19 | 777634.77 | 9715175.96 | 1436.38 | 304.09 | 319.61 | 339.58 | 320.30 | 327.17 | 329.51 | 332.09 | 326.89 | 325.01 | 312.37 | 324.38 | 323.69 | 5.68 |
| T20 | 777621.50 | 9715166.31 | 1435.48 | 304.72 | 322.90 | 337.56 | 323.41 | 320.83 | 330.54 | 334.72 | 324.28 | 323.11 | 320.14 | 329.19 | 315.00 | 9.62 |
| PI2 | 777666.36 | 9715141.99 | 1457.73 | 300.77 | 311.83 | 330.92 | 305.92 | 311.92 | 324.22 | 317.58 | 316.73 | 308.48 | 315.31 | 288.48 | 308.60 | 10.17 |

**Table A4.** The initial coordinates and azimuth of movement detected by 26 fixed points in Guarumales from 2014 to 2018.

| Point | Initial Coordinates (m) | | | Azimuth (Degrees) | | | | | |
|---|---|---|---|---|---|---|---|---|---|
| | x | y | z | 2014 | 2015 | 2016 | 2017 | 2018 | Average |
| PEG3 | 778008.63 | 9714864.70 | 1590.91 | 20.17 | 348.40 | 325.05 | 321.58 | 278.96 | 292.85 |
| PI3 | 778025.19 | 9715134.72 | 1556.52 | 9.09 | 348.33 | 329.56 | 328.89 | 269.35 | 297.50 |
| PI5 | 778251.24 | 9715176.83 | 1540.81 | 3.59 | 35.80 | 311.72 | 339.52 | 227.68 | 277.71 |
| PI6 | 778376.07 | 9715074.17 | 1609.79 | 17.30 | 351.38 | 325.82 | 330.01 | 263.06 | 286.85 |
| PI9 | 777522.50 | 9714794.21 | 1531.74 | 40.66 | 349.73 | 328.41 | 319.85 | 281.64 | 289.37 |
| PI10 | 777787.84 | 9715091.17 | 1501.97 | 15.03 | 347.44 | 332.73 | 326.52 | 264.19 | 290.94 |
| PI11 | 778141.91 | 9715116.04 | 1551.38 | 5.19 | 347.80 | 318.75 | 306.33 | 262.64 | 247.90 |
| PI12 | 778037.53 | 9714282.10 | 1786.93 | 15.20 | 351.03 | 342.58 | 338.65 | 322.19 | 297.94 |
| S1 | 778061.23 | 9715138.95 | 1556.09 | 14.45 | 349.79 | 332.69 | 333.84 | 270.06 | 279.40 |
| S2 | 778185.72 | 9715133.53 | 1550.76 | 4.53 | 346.01 | 320.47 | 336.37 | 238.71 | 261.07 |
| S3 | 778222.45 | 9715155.27 | 1546.36 | 1.90 | 346.44 | 315.68 | 339.57 | 232.32 | 257.70 |
| T9 | 777682.11 | 9714953.91 | 1512.26 | 27.23 | 347.22 | 329.09 | 318.07 | 274.22 | 287.23 |
| T10 | 777944.87 | 9714916.40 | 1572.02 | 23.49 | 348.90 | 330.70 | 328.17 | 276.87 | 289.82 |
| T11 | 778071.91 | 9715041.24 | 1585.34 | 14.87 | 349.86 | 335.80 | 340.60 | 266.98 | 293.65 |
| T12 | 778198.74 | 9714849.28 | 1654.86 | 28.70 | 357.12 | 321.72 | 331.22 | 278.25 | 288.39 |
| T13 | 778225.66 | 9714694.89 | 1678.90 | 32.60 | 4.24 | 313.58 | 332.86 | 288.72 | 274.19 |
| T14 | 778074.50 | 9714377.62 | 1752.03 | 27.05 | 0.17 | 334.72 | 333.52 | 313.74 | 276.42 |
| T4 | 778889.60 | 9714805.92 | 1823.11 | 56.90 | 35.23 | 151.16 | 156.55 | 271.41 | 135.62 |
| T8 | 777438.58 | 9714852.99 | 1500.13 | 41.68 | 347.90 | 336.15 | 306.31 | 278.69 | 281.92 |
| T18 | 778553.43 | 9714196.82 | 1892.91 | 37.98 | 341.75 | 303.14 | 300.44 | 311.62 | 276.82 |
| PI7 | 777610.35 | 9715060.41 | 1461.22 | 2.37 | 331.38 | 307.57 | 295.61 | 277.10 | 281.59 |
| T16 | 777671.99 | 9715309.82 | 1380.58 | 5.15 | 320.61 | 324.08 | 323.33 | 315.08 | 301.30 |
| T17 | 777524.25 | 9715149.42 | 1397.36 | 356.09 | 326.37 | 323.51 | 324.32 | 321.77 | 319.64 |
| T19 | 777634.77 | 9715175.96 | 1436.38 | 338.95 | 323.39 | 322.48 | 321.33 | 294.11 | 306.31 |
| T20 | 777621.50 | 9715166.31 | 1435.48 | 349.32 | 332.54 | 320.83 | 327.98 | 294.29 | 308.29 |
| PI2 | 777666.36 | 9715141.99 | 1457.73 | 344.86 | 316.39 | 311.01 | 310.52 | 277.41 | 296.56 |

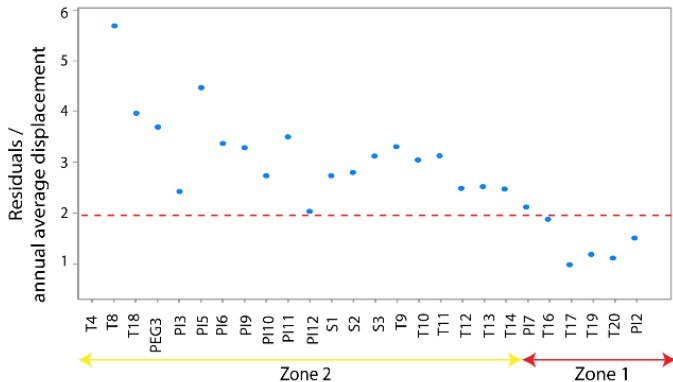

**Figure A5.** Residuals compared to the horizontal annual average displacement for the 26 geodetical observation points.

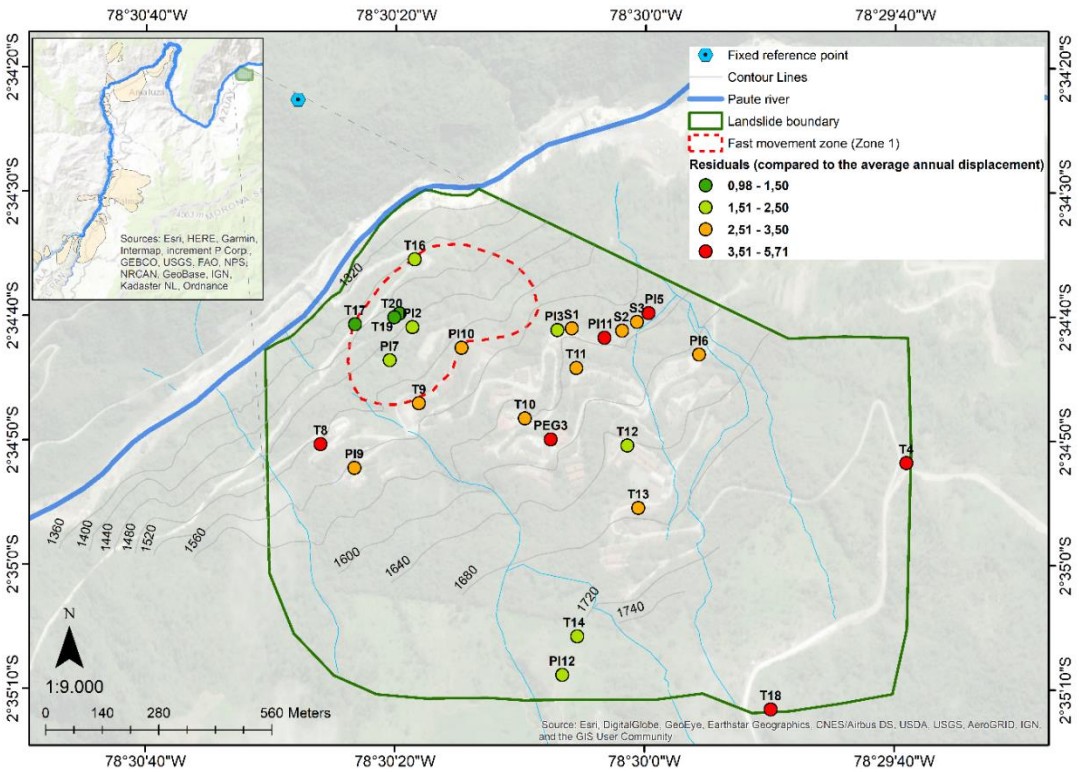

**Figure A6.** Plan view of the residuals compared to the annual average displacement in Guarumales.

Of the 11 piezometers only 3 had meaningful results in the time series analysis. Out of the 11, 10 showed significant fluctuations; however, those could not be linked via the (simple) time series analysis to meteorological forcing. However, the fluctuations were in the same order of magnitude.

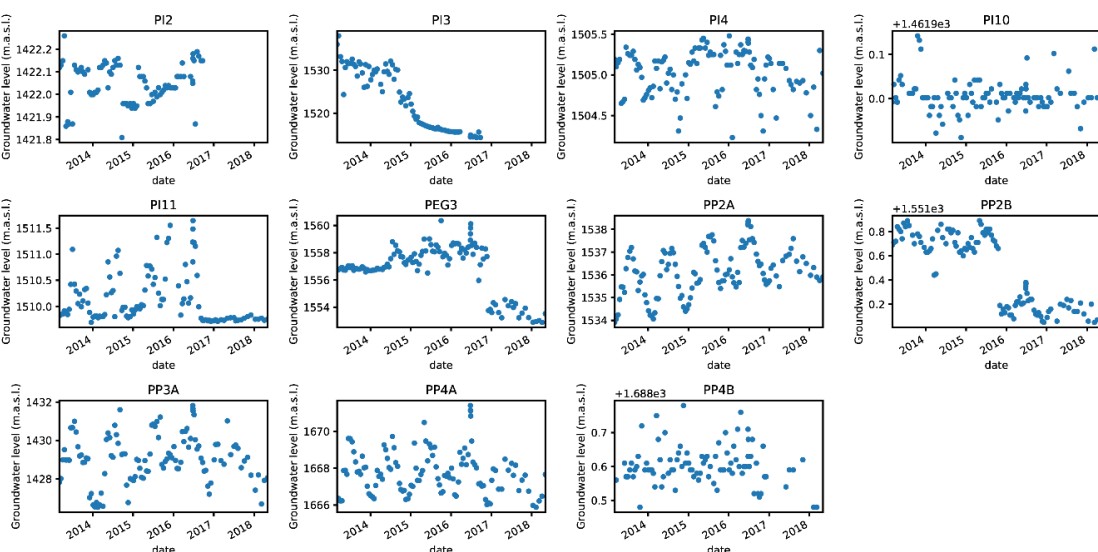

**Figure A7.** Time series plot of 11 groundwater observation points.

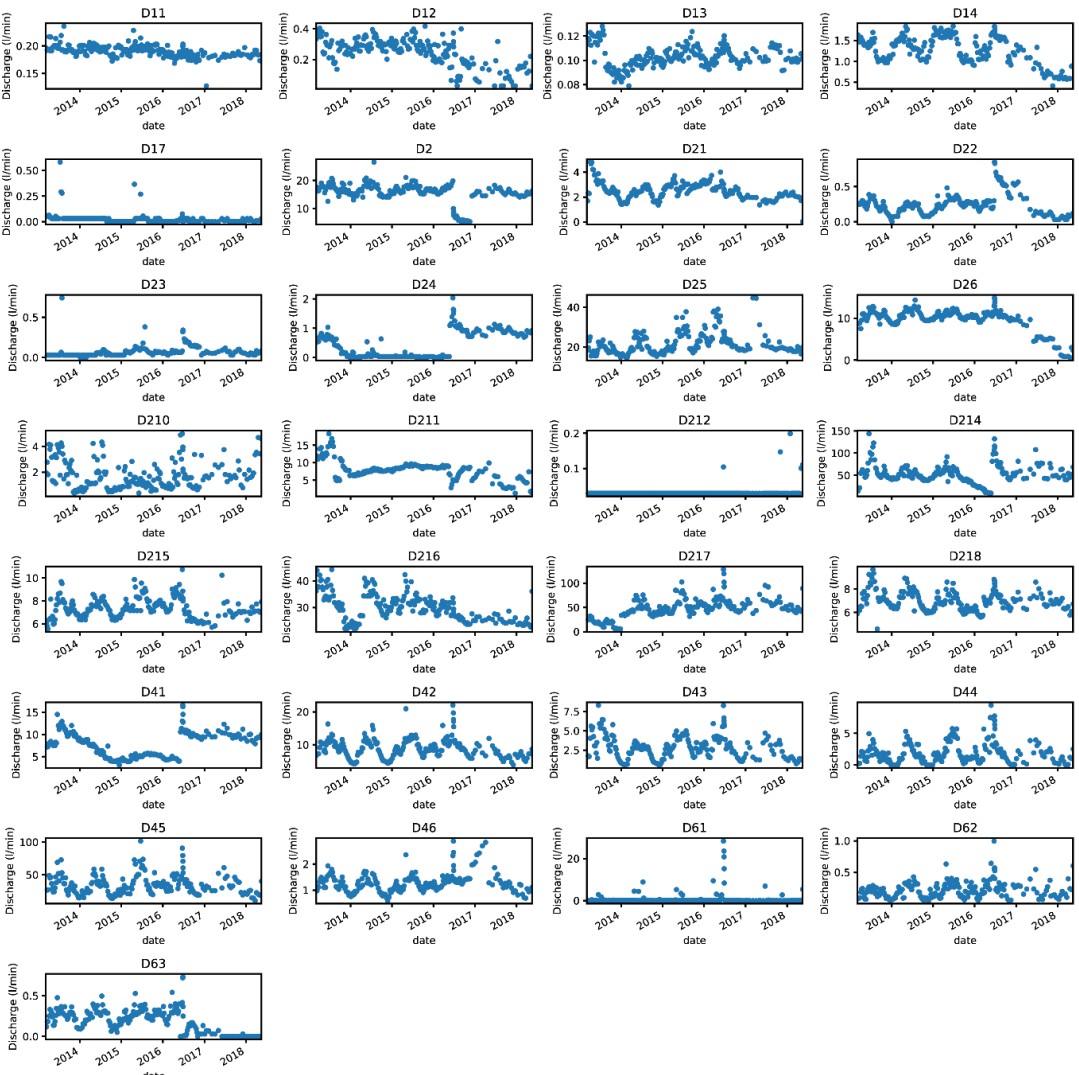

**Figure A8.** Time series plot of 29 monitored drains.

*Appendix A.6. Hydrogeological Classification for Landslides, from Brönnimann*

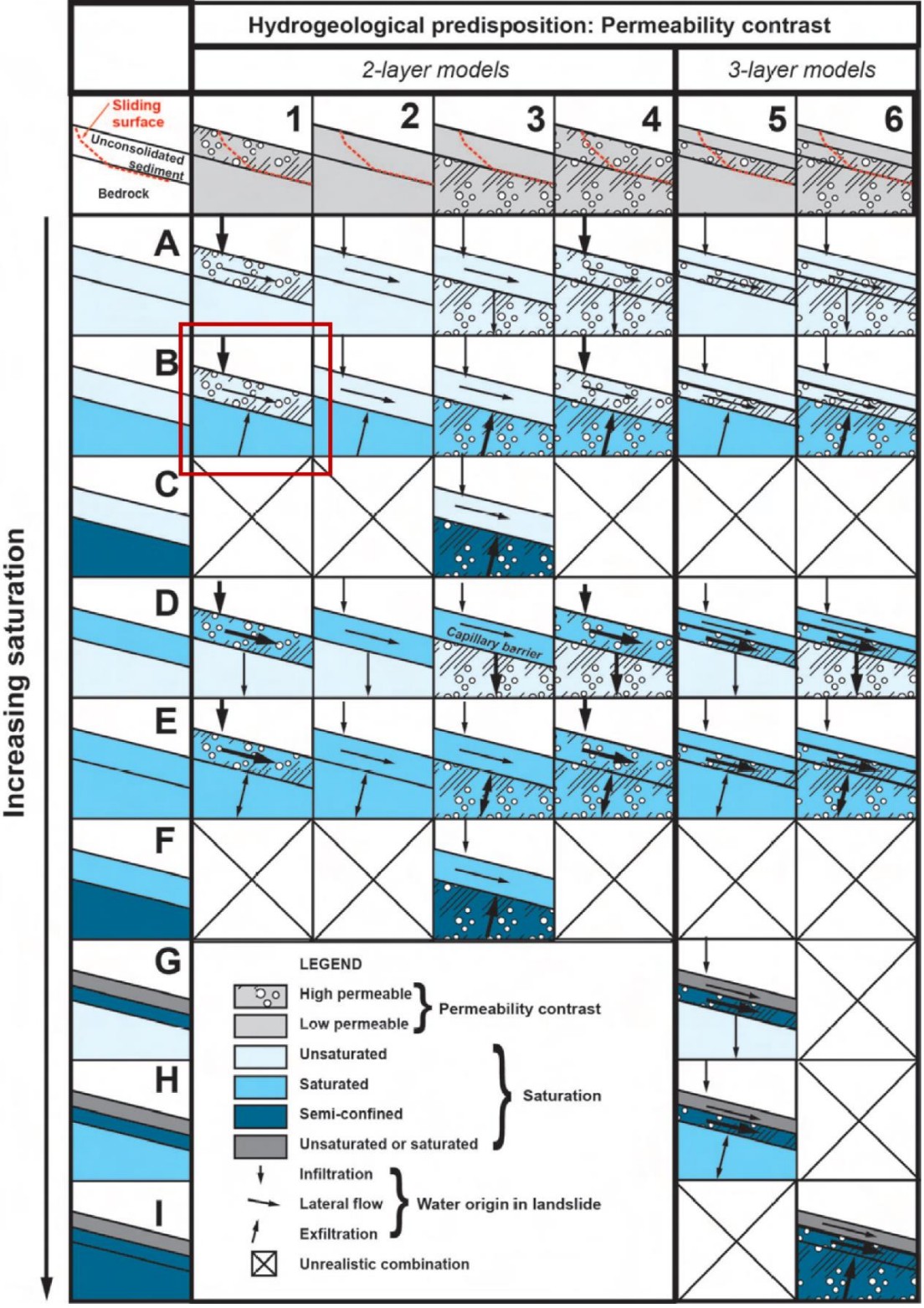

**Figure A9.** Hydrogeological classification for landslides, from Brönnimann [6]. Reproduced with permission from Laurent Tacher, Thesis: Effect of groundwater on landslide triggering; published by EPFL, 2011.

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
