# Peer review of "Characterization and Hydrological Analysis of the Guarumales Deep-Seated Landslide in the Tropical Ecuadorian Andes"

_geosciences, doi:10.3390/geosciences10070267_

Round 1
Reviewer 1 Report
Summary:
Vinueza et al. perform a detailed study of the Guarumales landslide, a large, deep-seated, slow-moving landslide located in Ecuador. Their primary objective is to investigate the predisposing factors and driving forces of the landslide. They examine 18 years of surface displacement data, 6 years of hydrologic data, and lithologic data to gain a better understanding of the controls on landslide motion. This landslide, and others nearby, are located near a major hydropower plant and thus constitute a major hazard.
Vinueza et al show the landslide exhibits a range of velocity spatially, typically between ~50-200 mm/yr, but overall, it moved at a fairly consistent rate over the 18-year period. They characterize the hydrologic response of the groundwater, and based on results of 3 wells, they find that the groundwater system responds to rainfall events within 10-30 days, and that groundwater changes can persist for 100-300 days after these events. However, they do not find a clear relationship between these short-term hydrologic changes and landslide motion which they attribute to the limited temporal resolution (yearly) and the accuracy (cm) of their displacement measurements.
Recommendation:
This study will be of interest to landslide scientists’ and hydrologists. It combines detailed hydrologic, kinematic, and geologic data for a deep-seated landslide which has high hazard potential. I think the manuscript is well written and provides an interesting dataset and with some additional work it will be ready for publication after MODERATE revisions.
Major Comments:
- I want to begin by saying I think this manuscript has the potential to be a significant contribution. It presents detailed hydrologic analysis that will be useful for understanding landslides. However, in its current state, the manuscript lacks key details that are needed to understand the findings. The Methods section is lacking the detailed descriptions necessary to understand the results and more could be done to connect the measured hydrology to the landslide mechanisms. I will elaborate more below.
- A primary result is that the water table stays tens of meters below the ground surface, close to the transition from the landslide colluvium to bedrock with only modest changes (relative to the landslide thickness) each year due to rainfall. While the authors are unable to make meaningful connections between landslide displacement and seasonal rainfall/groundwater changes due to limitations in the accuracy and sampling of the displacement data, what can they say about long-term ground water levels in relation to the steady creep motion displayed by the landslide? I feel there is an opportunity to expand this analysis and discuss how the relatively stable groundwater table may drive the stable landslide motion.
- To better understand the hydrologic response, more information about the wells is needed in the manuscript. At what depth are the wells drilled to? Why do most of the wells show complex behavior? I know there is mention of non-linear relationships, but since the majority of the wells do not show a simple behavior with rainfall, how should we think about the overall hydrologic response in governing the landslide behavior?
- The hydrologic time lag and memory are very interesting. First, as I will mention several times in this Review, I think a better explanation of the Pastas model is needed. I realize it is easy to look up the model in other papers, but a basic description is needed here to provide the reader enough information to understand how it works. Second, based on results from the Pastas model, the time lag for 3 wells ranges between 10-30 days. How deep are these wells? Are they sampling the same depth within the landslide? In other words, do the differences in depth below the ground surface explain the time lag changes. Third, the memory range between 100-250 days is also intriguing. Given it rains nearly year-round, with some changes in rainfall amount, does this suggest that these overlapping pore pressure pulses may be driving the landslide at steady creep rate?
- The Methods section is very brief and needs more detail so the reader can understand the authors approach. For instance, how were the rainfall and evaporation measured? Why were horizontal drains installed in the landslide in the first place (I assume to help drain it and slow it down, but this should be mentioned), how was the electrical conductivity measured and what does that data tell us? How does the Pastas package work? What are the assumptions? Importantly, I did not see any details about the borehole data or data used to interpret the thickness of the landslide.
- A main result is that the error is too large and the temporal sampling is too coarse to interpret interannual variation of displacement, however there is no error measurement shown in the Figures. Can the authors further explain the error calculations and add error bars to the displacement figures?
- Is there a concern that this landslide may fail catastrophically? If so, what may drive it to do so?
Minor comments
Line 23: I do not think the time-based precision of the total station is needed in the abstract. Please delete.
Line 25: change to “Our geologic investigation shows a locally complex…”
Line 26-30: Much of the abstract could be improved. Please use the active voice and use numbers to explain your results. For example, change to “We find that displacement rates are nearly constant at approximately 50 mm/yr over the 18 year study period, however the accuracy and time resolution of the displacement data did now allow us to resolve possible accelerations associated with hydro-meteorological forcing. We also found that groundwater and slope drainage showed a lagged response to rainfall ranging from 10 to 30 days”
Line 35: change “trigger” to “driver” and “displacement” to “movement”.
Line 43: change to “The occurrence of landslides is dynamic and is affected by time-dependent factors such as…”
Line 50: change to “require water to accumulate in the landslide body (either due to ”…
Line 60: The sentence beginning with “The most important historical cases” needs some clarification. What is important about these landslides? Did they cause the most destruction?
Line 67: Why is slope instability enhanced by the construction of artificial lakes? What about the lakes promotes landsliding?
Line 79: change “dragged” to “transported”
Line 82: change to “in the adjacent slopes accelerated or were reactivated”
Line 91-92: change to “(if any observed)”
Line 110-113: What is the vegetation like?
Line 122: delete “and” before “groundwater”
Line 124: change “unclear” to “unknown”
Line 131: change “data was” to “data were”
Line 142: change “Electric” to “Electrical” and “obtained” to “measured”
Line 158-159: This information should go along with the paragraph that discusses the electrical conductivity starting on line 142.
Line 160: Please define the Bronnimann classification system and how you use it. I realize there is a figure in the appendix (which should also be referenced here) but a description is also needed.
Line 167-170: This is a good example of a place where it’s unclear if the information is from this study or this is from previous work. Did this study do the geologic characterization? If so, how did you determine the rock type, composition, and age of the rocks? This information needs to be in the Methods section, unless it’s based on prior work in which case it needs to be defined with key refences.
Line 189: The Amaluza need a reference.
Line 196: Change to “represents the rest of the slope where the velocities range from 30 to …”
Line 205: Delete “displacement”
Line 206: Change “displacement” to “the velocity” and “indicating the behavior” to “which mostly corresponds to the behavior of zone 2”.
Line 208: delete “that go”
Line 208-209: The velocities ranged from 0 to 150.
Line 216-222: This paragraph would be better placed above in the Methods section. Also, is the trend line referring to the trend line in Figure 5b? Please specify.
Line 228: change “drain discharge increase” to “discharge from the drains increased”
Line 230: What is the depth of the well below the ground surface?
Line 230: change to “between when groundwater levels rise and rainfall occurs”
Line 238: change “with” to “from”
Line 244: Can you briefly explain the lack of relation between the rainfall and the other 8 piezometers?
Line 254-256: What is the depth of these wells? Are differences in lag time explained by different depths within the landslide?
Line 301-304: Please move the paragraph above where the displacement is mentioned.
Line 306: change to “in the slope that cause landslding”
Line 307: delete “if any observed”
Line 308-309: delete “and how this relationship affects slope stability”
Line 326-327: This sentence is already mentioned in the first paragraph of the Discussion. Combine with above.
Lines 334-337: These sentences can also be moved up into the first paragraph of the Discussion.
Line 337-339: The range in hydrologic response times is rather large. How should the reader interpret this when considering the landslide evolution?
Lines 359-363: The response time is discussed on lines 338 and 339. Can you combine these paragraphs?
Line 379: What is the context for the limited function? Do you mean that they do not work to stop the landslide from moving?
Line 384: delete “drain” before “12%”
Line 391: change to “allowed us”
Line 403: change “no change” to “no significant changes”
Line 412-413: what depth below the ground surface is the groundwater at these locations?
Figures:
Figure 1: The figure caption and legend could use some more information. Is the thick light blue polygon the lake from the dams? Add arrows on the Guarumales landslide to indicate direction of motion. Can you add a satellite photo too? This will help show what the surface conditions are like.
Figure 3: It is challenging to examine the cross-section Figure 3B with the variety of patterns. I suggest using some patterns and some colors to help differentiate the layers. Also, how much of this is interpreted based on extrapolation vs measured? My understanding is you have measurements from boreholes in certain locations and the rest must be interpretation?
Figure 4: Add error bars to Figure 4B. Add legends to the figures showing zone 1 and zone 2 with red and yellow color
Figure 6: Is there a reason the largest fluctuation in evaporation occurs in 2013? Please also add the depth of the well used for the groundwater measurement.
Figure 7: There appears to be a horizontal dashed yellow line on the time axis of both plots, but it’s not clear what that corresponds to, it’s also very hard to see.
Figure 8: Please add the electrical conductivity for rainfall to this figure.
Figure 9: The contrasting patterns make this figure hard to interpret. I suggest using some patterns and some solid colors to highlight different layers. The slip surface pattern in particular makes it hard to see the details. I suggest a solid line. Also, how well known is the slip surface depth? Presumably that is from the boreholes and then interpreted in other places?
Figure A1-A2: Use combo of colors and patterns instead of all patterns. Hard to decipher each layer as is.
Appendix D: Can you highlight the classification model you think represents your landslide?
Reviewer 2 Report
I have read and considered the manuscript presented by Urgilez Vinueza and colleagues. The authors present a study of the Guarumales landslide, a large deep-seated and slow-moving landslide in the Eastern Andes of Ecuador. They aim at analysing the predisposing and triggering factors affecting the overall landslide stability, building on long time series of measures (including surface displacement and piezometers). The hydrology and broad mechanisms of the landslide are described and used to develop a conceptual hydrological model.
This manuscript proposes an interesting study of a deep-seated landslide in the too-often overlooked tropical environmental context. Long-term and detailed hydrological and displacement measurement provide a rare (but essential) view on the hydrology of such landslide, hydrology itself being key for understanding the overall landslide stability and predicting future landslide displacements. This manuscript is worth publishing in Geosciences, but several points that I develop hereunder need first to be addressed. I, therefore, suggest that the manuscript be subjected to moderate revisions before publication.
I hereafter provide some general recommendations:
- This article would firstly benefit from a reorganisation of the section. Within sections, a lot of ‘zigzag’ are made between subjects and results and observations are too dispatched over different sections, making difficult for the reader to follow the authors’ case at first instance. This is a major flaw, but which could be rapidly rectified by following a stricter distinction between the different method/result/discussion sections. The readability of the manuscript would greatly benefit from such editing. Note also that some sentences are excessively long and may lose the reader. I have provided some hints for the reformulation of some hereunder.
- This is a rare detailed study (in the English-speaking scientific literature at least) of large slow-moving landslide in tropical environment. Could these environmental conditions affect the causes and triggers of landsliding? Here are some references that the authors may found interesting regarding these questions: (Thomas, 1994; Gupta, 2011; Dille et al., 2019).
- Some interesting insights on landslide groundwater systems could also be learned from this article: Belle et al., 2018 Control of Tropical Landcover and Soil Properties on Landslides’ Aquifer Recharge, Piezometry and Dynamics
Some specific comments regarding the different sections are provided hereafter (note that some comments are also provided directly in the pdf):
- Abstract:
- Zigzag between ideas and observations are found in most sections, including the abstract. I here propose a formulation of the abstract that allows removing most of them. Note that it is only a suggestion.
à High landslide potential along the steep hillslopes of the Eastern Andes in Ecuador provides challenges for hazard mitigation, especially in areas with hydropower dams and reservoirs. The objective of this study is to characterize, understand, and quantify mechanisms driving motions of the Guarumales landslide. This 1.5 km² (?) deep-seated, slow-moving landslide is actively moving and threatening the “Paute Integral” hydroelectric complex. Building on long time series of measurements of surface displacement, precipitation, and groundwater level fluctuations, we analysed the role of predisposing conditions and triggering factors on the stability of the landslide. We performed an analysis of the time-series of measured groundwater levels and drainage data using transfer functions. The geological interpretation of the landslide was further revised based on twelve new drillings. It shows a locally complex system of colluvium deposits overlying a schist bedrock, reaching up to 100 m. Measured displacement rates are nearly constant over the 18 years of study. However, measurement accuracy and time resolution were probably too small to identify possible acceleration or deceleration phases in response to hydro-meteorological forcing. Groundwater and slope drainage showed a lagged response to rainfall. Finally, we developed a conceptual model of the Guarumales landslide, which we hope will improve our understanding of the other many deep-seated landslides present in the Eastern Andes.
- Note that keywords are made to provide additional words over the one found in the title for helping indexation in search engine. Selecting similar words in the title and the Keyword section has therefore no sense.
- Introduction
- I would be glad to have a paragraph describing what research has already been conducted on the landslide, in particular considering that the referred literature is usually Spanish. If relevant English-speaking reference(s) are available on landslide processes in Ecuador I would suggest including it/them.
- How is the landslide threatening the hydropower activities?
- Section “Description of the study area”
- I strongly suggest restructuring this section, e.g., to avoid zigzag between different scales (e.g., should start from the lowest scale (location within Ecuador) to the higher scale (the landslide); à location within Ecuador à geology of Ecuador (but should be limited to essential information to understand landsliding processes) à Paute River basin (and climate) à Paute Integral hydropower plant à the landslide)
- What is the elevation of the landslide?
- Consider adding a picture (or satellite image) of the landslide so that the reader can better apprehend the process at play.
- Methods:
- Again, there are too many zigzags. First discuss (for e.g.) the measure of the surface displacements (period, number of measures, accuracy, etc), then rainfall, then groundwater,…
- I would personally try to reduce the number of acronyms and numbering of sites for drains, piezometers, etc. It makes the paragraphs confusing for the reader while this information is not essential (could be summarized and put in appendix).
- Why were a) drains installed in 1994 and b) electric conductivity tests conducted could be respectively explained in one sentence.
- Description of the model used for the simulation of groundwater level is absent (what model? What parameters? Simple linear diffusion? What soil permeability/diffusivity parameters are used? Is the model used in the literature for landslides? What are the differences with e.g. simple models developed by (Handwerger et al., 2013, 2019)).
- Why the authors decided to run simulations while direct measurements were available from 11 piezometers should also be stated clearly.
- What about the drilling campaigns of 2016 and 2019? What was their depth? Did they allow to estimate the depth of the slip surface?
- Results:
- Some paragraphs actually belong to the methods.
- More information could be shared about results from the drilling campaigns of 2016 and 2019. E.g., what about the weathering intensities, depth (considering tropical environments)?
- Only yearly velocities are shown, while you have 8 measurements per year, why? This is an important question also regarding the discussion.
- I’m not asking to do that in this manuscript, but is there any other measurements available (e.g., from continuous dGNSS, Remote Sensing)?
- 6: Why is the piezometric level higher during the dry season than the wet season? (e.g., in 2015, 2016, 2017)?
- L133 it is stated that " Groundwater data were collected using a manual water level meter two times per month. " If groundwater levels are measured twice a month, how are measured peak response of the unconfined groundwater system to rainfall of ~5, 10, 30 days?
- Section 5 “Hydrogeological conceptual model of Guarumales slope”:
- I must admit that I do not follow how the authors reach such conclusions about the most adequate model to explain the landslide hydrology. This section would benefit from some editing to make the arguments clearer.
- Discussion:
- This section is (again) difficult to follow because of the zigzag between paragraphs. I recommend to a) merge discussion on landslide velocity together (L309-313 and L326-358). b) start the discussion on the groundwater system by discussing the observed values (L373-378) before discussing of simulation results. c) It would be great to avoid short (e.g., 3-5 lines only) paragraphs.
- A large body of literature exists on the influence of groundwater fluctuations on landslide stability. The discussion of the process at Guarumales landslide would greatly benefit from including (and discussing) more based on those references (I provide a far-from-exhaustive-list here: (Iverson, 2000; Schulz et al., 2009; Van Asch et al., 2009; Handwerger et al., 2013; Van Der Spek et al., 2013; Bogaard and Greco, 2016; Carey et al., 2019)).
- What does past inclinometer data say about the landslide rates of movement?
- What about using dGNSS measure or remote sensing to measure surface displacement rates with a much spatio-temporal accuracy? Has it been done already?
- The potential influence of evaporation (and evapotranspiration) is not discussed. In another tropical setting, Belle et al. 2018 showed that a large quantity of the rainwater was absorbed by the upper soil layer, this reservoir itself subjected to high real evapotranspiration (1500 mm/year) due to the dense tropical broad-leaved vegetation.
- Faster displacements are observed over some zones. Could you discuss the presence of different kinematic elements within the large landslide complex? (“Landslides often comprise different kinematic elements, and conditions affecting their movement, such as material properties and porewater pressures, vary in time and space.” (Schulz et al., 2017))
- What about future landslide behaviour? How is it threatening the hydropower plant?
- Conclusion:
- A conclusion generally ends with a sentence putting the work in a broader context.
- “Pore water fluctuations are too small to have a significant effect on the landslide movement.” Are they? Can this be stated while no e.g. daily, weekly, monthly displacement measurements are available?
- Regarding the figures:
- Figure 2: I suggest using a hillshade of a DEM as figure background so that the reader can better grasp the extent and morphometry of the landslide
Some minor issues in the text:
With “à” I suggest a new formulation
- L34: “movement of mass” What mass? Precise
- L35: gravity is not the main trigger of displacement. It is however indeed the main driver of displacement (see Sidle and Ochiai 2006)
- L39: predisposing conditions of landslides à predisposing conditions for landslide occurence
- L43: is it the occurrence of landslide that is a dynamic process?
- L52-53: is there always a hydrological threshold? Probably not. I would rephrase this sentence
- L57: situated à sited
- L58: It has been the scene of many landslides à “It is a highly landslide-prone region, with landslides affecting Ecuadorian landscape, society, and economy “. I also suggest putting at least one reference in English.
- L60-61: The most important historical cases of mass movements in Ecuador are related to à Some key reported landslides in Ecuadorian Andes include e.g., …
- L62-65: already stated in L46-49
- L66-68: à At the transition between the Andes and the low-lying Amazon rainforest (?), landsliding is further enhanced by the construction of artificial lakes created for hydropower production. Reference?
- L68-71: àAbout 35% Ecuador’s energy (electricity?) is generated from three dams forming the Paute Integral hydroelectric complex, over the Paute River. Along those reservoirs have been identified Twenty-on deep-seated landslide. Reference?
- L74-86: à An example is the La Josephina landslide, which occurred on the 29th of March 1993 at the junction between the Paute and Jadán rivers (REF). Its catastrophic failure provoked the destabilisation of the other hillside, the Tamuga hill, killing more than 100 people. These two landslides blocked the rivers, leading to the upslope flooding of approximately 1000 ha. It later favoured the formation of flash floods events, causing riverbanks erosion and the transport of a large amount of sediment downslope [18–21]. Following these events, many landslides of the adjacent slope seem to have been accelerated or reactivated (REF). One of these is the deep-seated and actively moving Guarumales landslide. ++
- L98: Real mountain range à Cordillera Real?
- L97: add lon/lat coordinates (à The Guarumales landslide (2.57 S, 78.5 W), is a X km² deep-seated landslide located in south-east Ecuador.) Elevation of head and toe?
- L103: public company
- L107-108: I would consider removing the sentence: Temperature and humidity vary gradually with height and cause stratification of vegetation [22].
- L108-109: I don’t understand this sentence.
- L126: consider using surface movement instead of surficial movement (this suggestion is valid for the entire manuscript)
- L127: “The outcome of this survey” à consider rephrasing
- L133-135: à Groundwater level data were collected manually at 11 piezometers twice per month.
- L136: à 36 horizontal drains were installed within the landslide since 1996. Out of them, 26 are operational and forms four groups over the landslide.
- L142: Explain in one sentence why electric conductivity measures were done
- L152-154: à We analysed how groundwater levels and drain discharges are affected by rainfall and reference evaporation [27], using transfer function noise (TFN) modeling implemented in the Python package Pastas [28].
- L219: What does Zone 2 represent? All the landslide except zone 1? It should be made clearer.
- L220: “Residuals for zones 1 and 2 are ~1.7 and ~3.5 (up to 6) times the average yearly displacement, respectively.” Those are really large errors. Where do they come from?
- L229-230: à A time lag is always observed between groundwater levels rise (here for PP4A), and rainfall (Figure 6d).
- L236-237: remove ‘using time series analysis’
- L237-242: This is part of the method.
- L246: Would you expect another result? What explain that you obtain such relationship for only 3 out of 11 piezometer data? (for discussion)
- L279: I would personally refer to it as “Conceptual hydrological model of Guarumales landslide”.
- L306-309: à In this study, we aimed at analysing the role of predisposing conditions and triggering factors on the stability of the Guarumales landslide.
- L309-313: à 18 years of displacement measurements showed that yearly surface displacement rates are constant over time. While rainfalls have been shown to strongly influence the unconfined groundwater system with a ~10-30 days response time, no monthly variation in displacement rates were observed with within our displacement data.
- L339-340: Isn't it more 'geodetical accuracy is low compared to...' ?
- L343-345: It lacks a reference. And why RTK? It has lower positioning accuracies over PPK dGNSS... What about remote sensing? InSAR ? (but displacement seems quite northward)
- L354: Consider naming it dGNSS instead of GPS.
- L348-351: How would seismicity explain a different displacement direction?
- L353-356: This is interesting and could be used in a discussion of the future stability of the slope.
- L359: State explicitly why you simulate your groundwater levels while you have direct measurements from 11 piezometers.
- L379-381: This sentence is complex to read and should be rewrite.
- L392-394: This sentence should be checked.
- L399: the possible driving mechanisms underlying the Guarumales landslide à the mechanisms interfering with the stability of the Guarumales landslide
- L401: remove “subsequent”
- L407:à A conceptual hydrological model was developed for Guarumales landslide.
- L408: “It reflects the current state of knowledge of the landslide.” can be removed
- L413-414: “Pore water fluctuations are too small to have a significant effect on the landslide movement.” Are they? Can this be stated while no e.g. daily, weekly, monthly displacement measurements are available?
- L418-420: à “This encompass the collection of displacement and groundwater level data with a higher spatio-temporal accuracy and resolution.” A lead to obtain such data could be added (e.g., using dGNSS acquisitions, SAR interferometry, etc.).
- L420-421: The sentence “The resolution of the groundwater data must also be increased to once a week, to match the resolution of the displacement data.” Can be deleted.
References cited in the review:
Belle, P., Aunay, B., Lachassagne, P., Ladouche, B., Join, J.L., 2018. Control of tropical landcover and soil properties on landslides’ aquifer recharge, piezometry and dynamics. Water (Switzerland) 10, 12–14. https://doi.org/10.3390/w10101491
Bogaard, T.A., Greco, R., 2016. Landslide hydrology: from hydrology to pore pressure. Wiley Interdiscip. Rev. Water 3, 439–459. https://doi.org/10.1002/wat2.1126
Carey, J.M., Massey, C.I., Lyndsell, B., Petley, D.N., 2019. Displacement mechanisms of slow-moving landslides in response to changes in porewater pressure and dynamic stress. Earth Surf. Dyn. 7, 707–722. https://doi.org/10.5194/esurf-7-707-2019
Dille, A., Kervyn, F., Mugaruka Bibentyo, T., Delvaux, D., Ganza, G.B., Ilombe Mawe, G., Kalikone Buzera, C., Safari Nakito, E., Moeyersons, J., Monsieurs, E., Nzolang, C., Smets, B., Kervyn, M., Dewitte, O., 2019. Causes and triggers of deep-seated hillslope instability in the tropics – Insights from a 60-year record of Ikoma landslide (DR Congo). Geomorphology 345. https://doi.org/10.1016/j.geomorph.2019.106835
Gupta, A., 2011. Tropical Geomorphology. Cambridge University Press.
Handwerger, A.L., Huang, M.-H., Fielding, E.J., Booth, A.M., Bürgmann, R., 2019. A shift from drought to extreme rainfall drives a stable landslide to catastrophic failure. Sci. Rep. 9, 1–12. https://doi.org/10.1038/s41598-018-38300-0
Handwerger, A.L., Roering, J.J., Schmidt, D.A., 2013. Controls on the seasonal deformation of slow-moving landslides. Earth Planet. Sci. Lett. 377–378, 239–247. https://doi.org/10.1016/j.epsl.2013.06.047
Iverson, R.M., 2000. Landslide triggering by rain infiltration. Water Resour. Res. 36, 1897–1910. https://doi.org/10.1029/2000WR900090
Schulz, W.H., Coe, J.A., Ricci, P.P., Smoczyk, G.M., Shurtleff, B.L., Panosky, J., 2017. Landslide kinematics and their potential controls from hourly to decadal timescales: Insights from integrating ground-based InSAR measurements with structural maps and long-term monitoring data. Geomorphology 285, 121–136. https://doi.org/10.1016/j.geomorph.2017.02.011
Schulz, W.H., McKenna, J.P., Kibler, J.D., Biavati, G., 2009. Relations between hydrology and velocity of a continuously moving landslide-evidence of pore-pressure feedback regulating landslide motion? Landslides 6, 181–190. https://doi.org/10.1007/s10346-009-0157-4
Thomas, M.F., 1994. Geomorphology in the tropics: a study of weathering and denudation in low latitudes. John Wiley & Sons.
Van Asch, T.W.J., Malet, J.P., Bogaard, T.A., 2009. The effect of groundwater fluctuations on the velocity pattern of slow-moving landslides. Nat. Hazards Earth Syst. Sci. 9, 739–749. https://doi.org/10.5194/nhess-9-739-2009
Van Der Spek, J.E., Bogaard, T.A., Bakker, M., 2013. Characterization of groundwater dynamics in landslides in varved clays. Hydrol. Earth Syst. Sci. 17, 2171–2183. https://doi.org/10.5194/hess-17-2171-2013

Reviewer 3 Report
The paper describes geotechnical aspects of a large deep-seated landslide located in the Eastern Andes of Ecuador. The authors assess the predisposing and triggering factors of slope movement based on data on hydrometeorology, geology and groundwater levels. Based on the observations, they propose a model for slope movement.
The paper nicely documents the potential risk of such deep-seated landslides for hydro-electrical power plants and associated infrastructure.
In my opinion, the following issues merit further attention.
(1) The introduction of the paper is a mix of very general landslide definitions and a description of very specific landslide events in the Ecuadorian Andes. The contextualisation of the work is a bit missing, and it is not clear which research questions are addressed in the paper. In my view, the hydrogeological predisposing and triggering factors of landslides have not been studied in detail in the region, and this could be one new way of focusing the paper. This would also allow the authors to move the 2nd part of the introduction that corresponds with a description of landslides in the area (L57-65, L74-88) to the description of the study area.
(2) The authors refer to the geomorphological setting of the Andes Cordillera, with its predisposing factors for landslide hazards. There exist peer-reviewed literature on the predisposing and triggering factors of landslides for the Ecuadorian Andes. Also, for the specific region under study, there exists literature on the role of topography, geology and land use on landslide occurrences. The authors might want to revise the current state of the literature, to improve section L57-65.
(3) The 2nd part of the introduction contains a lot of detail that is difficult to grasp without having read the material on the study area. I would move these sections to the "study area". Also, not sure how relevant the description of the "La Josefina" landslide is for your study. What is the evidence that there is a link between "La Josefina" and "Guarumales", that are separated by about 70-80km(?).
(4) The description of the study area is hard to read, as the authors jump a bit from information on the Guarumales landslide, to the "study area" and to the "Paute basin". I would suggest that the description of the study area starts with an overall description of the location of the area within Ecuador, and within the Paute basin. Then, the location of the infrastructure and landslides from Figure 1. And finally, more details on the Gauramales LS, and the study area.
(5) The geology of the area needs more attention. The authors state that it is complex - which might be true, but I would like to know the main lithological units, their age, and their geotechnical properties (shear strength, internal angle of friction). A geological map is missing - a figure with location of the main tectono-lithological units is crucial.
(6) The methodology resumes the data that are available. This section would need a paragraph that states the rationale that was used to collect specific datasets. How are piezometers distributed, and why? Why did you consider only the geodetic points that are located on the landslide surface? How can you measure deformation when taking measures on the sliding body (only)? Where is the rainfall and meteo data coming from (name of the station, its location - altitude)? What is the idea behind collecting electrical conductivity measures? In how far EC is affected by temperature, DO, and anthropogenic waste?
(7) The part on the geology (4.1) is now written as part of the results. I would suggest to move it to the description of the study area, as none of the datasets or methods described in your section 3 provides information for what is presented in 4.1 geology. If you want to keep it in the "results", you would need to give more data on the geological boreholes, and eventually the geophysical studies that led to Figure 3 and the figures presented in the Appendix A. Also, the geology needs some attention. I do not see how an intrusive body can be described as a "intrusive deposit". It is either an igneous rock, and then it is "uncommon" to find it in the midst of a colluvial package. Or it is a "deposit or layer" composed of granodioritic material, but then it should be clarified how it got there - by slope processes? from where?. On L189, the authors state that the granodioritic body has an outcrop along the Paute River. Where can I see this on Figure 3?
(8) It is not entirely clear how the surface displacement vectors were calculated. Are the values presented in section 4.2 the absolute displacement values (projected on horizontal surface)? Also, when accumulating values at longer time intervals, how do you account for changes in the direction of movement? Is the cumulative value representative for the TOTAL movement in the (X,Y) direction? So, some of the point had a displacement of > 2m in the horizontal direction. What is the accuracy of the individual measurements? And why not taking a reference point outside the sliding area?
(9) The analysis of the hydrometeorological data is not very clear. The results are now focused on the model results of part of the dataset. In total, there are 11 piezometers for which data are available. If I read it well (L244-245), only 3 of them gave reasonable results. What about the others? Is there a systematic difference in the piezometer levels, and some correlation with topography? or lithology? Also, I would expect to some kind of correlation between the piezometer PP-2(A-B) and Group 4 (based on their location)? Is this the case?
(10) The last part of the results (hydrogeological model) and the discussion is well written. It clarifies several questions that I had when reading through the previous sections. Given the accuracy of the displacement measures, it is relevant to have error bars on the graphs that were presented previously. Also, the setup of the monitoring campaign needs some clarification (i.e. depth of the drains).
The authors make a lot of reference to the grey literature in Spanish. A quick count shows that about 50% of the references are referring to material in Spanish from local studies or reports (informes) that are not available for the general public. Good practice is to mention the essential documents, and refer to original research articles from peer-reviewed studies that are relevant for your study, and that can be found in the scientific literature by your readers. Below, I mention a few articles that might be helpful.
and finally, the language needs to be revised. There are several sentences that do not read very well. It is sometimes hard to read through the text because of the structure of the sentences
Minor editorial comments
English spelling and grammar needs to be checked. There is some odd wording in the text, probably from translating spanish terms to english.
L14: "the potential for landslides" - Do you mean landslide hazard or risk? Landslide susceptibility?
L14: "high sloped hills" - steep hill slopes?
L20 & L53: "slope acceleration" - do you mean "increased slope movement"?
L21: "Surficial displacement" - "displacement of the ground surface"?
L23: What do you mean with "6 and 5 seconds of precision"? Can you give accuracy and precision in cm or mm?
L25: "geology...locally complex system of colluvium deposits overlying a schist bedrock" - can you be more precise. As I read it now, you have two packages - colluvium and schist bedrock. What do you mean with "locally complex system"?
L29-30: In how far are geotechnical studies site-specific? In how far can you extrapolate the conceptual model of Guarumales to other landslides in the area? This might be highly depend on the geology, which can be highly variable in active tectonic settings such as the Eastern Cordillera of the Andes.
L40: rephrase "(geological) slope layers and the degree of saturation". Do you mean the local geology? What do you mean with "geological slope layers"?
L42: "different landslides" => do you mean different landslide types? different landslide magnitude? Please clarify
L47-48: Precise what you mean with "current, and evolving soil parameters within the landslide". Very often, landslides extend to the weathered regolith, and are (much) deeper than the soil material. I guess you want to include the weathered material on top of the hard rock as well - not only "soil".
L50-51: The paragraph starting on L50-51 is not well structured. From the 1st sentence, one gets the idea that deep-seated landslides are only possible when there are "accumulated conditions of water content". What about the role of seismicity?
L64-65: Here, you mention as triggering factors: rainfall and seismicity. In the paragraph on L50-L56, you state that deep-seated landslides require "accumulated conditions of water content", with no mention about seismicity. The two paragraphs needs to be revised so that there is a logical flow of thoughts.
L58-60: There has been a lot of work on landslide occurrences in the Andes. A good practise is to refer to the latest state of the knowledge on the topic. What have other researchers found? What are the regions that are most affected by landslides? (the statement that LS are concentrated in the Andean and sub-Andean regions is very broad...)
L62-65: Also, on the predisposing factors, try to be more specific. What is published in the scientific literature on the effect of geomorphology, anthropogenic impacts etc. on landslide occurrence?
L66-67: The authors state here that "slope instability is enhanced by the construction of artificial lakes created for hydropower production". This is a very strong statement, and needs to be underbuilt with scientific evidence. Otherwise, it is a hypothesis, and needs to be formulated as such.
L97-98: Rephrase, sentence is not clear. How can the Guaramales landslide extend through the Eastern Andes?
L98: Check the wording. Is the Eastern Andes the same thing as the Cordillera Real?
L100-101: From the map, it seems that Molino is on the other side of the river valley. How is Molino affected by the landslide?
L114-116: This sentence on the geology is not very informative. I would like to know the main units, their lithology and age, and their geomechanical properties. Ideally, a new figure with a good geological map, and indication of faults and main tectonic-lithological units. What do you mean with "shale and schist facies"?
L148-151: This section is not clear. What is the accuracy of the measures? How do you propagate errors from measurement errors? Where is the reference point to determine "movement" if all points are part of a moving surface?
L188-189: Quite particular to find a "intrusive deposit". Please check if this is possible at all from a geological/geomorphological point of view.
L197: In the appendix, you give measures of displacement. Can you clarify if these measures are TOTAL displacement values (X,Y,Z) or are horizontal displacement values (X,Y)? Also, are the displacements always in the direction? I would expect that there is some variability in the direction of the movement over time, with bulging and subsidence taking place as a function of the groundwater and saturation of the colluvium.
L202-203: It would be very helpful to have a map of the landslides, with indication of the direction and magnitude of movement for the geodetic point. You could do this by using arrows, where the direction is the azimuth, and the thickness or length of the arrow is the magnitude.
L273-276: Not sure what is the relevance of using electrical conductivity data here. Do you want to use mixing models to see the origin of the water drained? There might be a lot of other variables influencing water conductivity. Not sure how it contributes to make your point on the hydrogeological classification.
L280: What do yo mean with "local complex geological setting"?
Figure 1: Can you indicate in the legend what the red circles mean?
Figure 3: The geological profiles are very interesting, but hard to read. Can you put the different layers in colours (the shades do not reproduce well in a pdf)?
Figure 4: For the cumulative horizontal displacement, how do you account for changes in the direction of the movement?
Figure 5b: The figure is not clear. Not sure what is plotted here. Can you use maps of the landslide area, and show arrows indicating magnitude and direction of movement instead (and maybe, indicate the difference in magnitude in 2013)?
Figure 6: Puzzling figure. From figure 6, one could say that the total discharge from the drains is quite well correlated with the rainfall, and that the piezometer level (at least in PP4A) show a very poor correlation with rainfall or has a very long lag time. Do you see this for all piezometers, independent of their location along the slope? How deep are the drains inserted in the colluvium?
Figure 9: Good figure, but where you situate now the "intrusive deposits" in this model? Also, from Figure 1, it seems that the Guarumales landslide is located on the foot slopes of a much longer (very steep) hillslope. Is this the case?
References
Guns, M., Vanacker, V., 2013. Forest cover change trajectories and their impact on landslide occurrence in the tropical Andes. Environ. Earth Sci. 70, 2941–2952.
Muenchow, J., Brenning, A., and Richter, M.: Geomorphic process rates of landslides along a humidity gradient in the tropical Andes, Geomorphology, 139, 271–284, 2012.
Riemer, W., Locher, T., and Nunez, I. (1988). Mechanics of deep seated mass movements in metamorphic rocks of the Ecuadorian Andes. In Proceedings of the 5th International Symposium on Landslides, Lausanne. Edited by Ch. Bonnard. A.A. Balkema, Rotterdam. Vol. 1, pp. 307-310.
Tibaldi, A., Ferrari, L., Pasquarè, G., 1995. Landslides triggered by earthquakes and their relations with faults and mountain slope geometry: an example from Ecuador. Geomorphology 11(3), 215-226.
https://doi.org/10.1016/0169-555X(94)00060-5.
Vanacker, V., von Blanckenburg, F., Govers, G., Campforts, B., Molina, A., Ku-bik, P., 2015. Transient river response, captured by the channel steepness and its concavity. Geomorphology228, 234–243. http://dx.doi.org/10.1016/j.geomorph.2014.09.013.
Round 2
Reviewer 3 Report
The authors have revised their work thereby addressing some of the comments that were raised in the previous review round. The authors have revised the text, and have improved language and structure.
However, in my view, the manuscript needs a more thorough revision of the material presented, the interpretation of results and methodological setup. These comments were merely addressed, and I come back to them below. In my view, the following issues need to be addressed adequately to get the paper in a format ready for publication in a peer-reviewed international journal.
(1) The study focuses very strongly on a case-study located in the Ecuadorian Andes. There needs to be a transition between the very general statements at the beginning of the introduction (based on knowledge from reference manuals and textbooks in landslide analyses) and the specific description of landslide characteristics in the remaining part of the introduction.
When referring to landslides in South America, and the geo-dynamic setting of Ecuador, the authors base their arguments on grey literature - unpublished and unreviewed material in Spanish that is not available for the reviewers or readers of the journal (reference 15-17, 23-26, 29-31, etc). Two reviewers already highlighted that references in English need to be provided when they are relevant and available.
For the “geo-dynamic setting of Ecuador” and the occurrence of landslides on the Andean escarpments, there are plenty of studies on the geomorphology and distribution of landslides in Ecuador : (Muenchow et al., 2012; Tibaldi et al., 1995; Coltorti and Ollier, 2000; Guthrie and Evans, 2007; Baize et al., 2015). For an international peer-reviewed article, it is important to embed the study in the latest state of the art in the international literature.
(2) As mentioned in the previous review round, it is not clear why the Josefina landslides is described in much detail in the introduction. The authors give some very general information on the landslide characteristics. For the readers, the link between Josefina and Guarumales is not clear. I would only keep this paragraph when (a) you can indicate based on peer-review literature what the Josefina landslide is telling us about the dynamics of deep-seated landslides in the region, and/or (b) you can indicate that there is a measurable impact of the flood of the Josefina breakthrough and the Guarumales landslide based on peer-reviewed literature and evidence-based material.
(3) In the text, the authors refer to different locations : Josefina, Guaraumes, Paute, Paute basin, Cuenca. We need a good map where the location of these points is shown.
(4) Understanding the geology is crucial for reconstructing landslide dynamics. The description of the geology needs to be revised based on recent literature (see work by Spikings et al, 2001; and more recent work by Pratt et al., 2005). The description of geology is just not correct - the Alao-Paute Unit is part of the Alao-Paute terrane, and composed of metamorphic rocks. What do you mean with “insular arc”? What the source of your data that says that the Alao-Paute unit contains tuffs and volcanic agglomerates? Also, what about the igneous block in your landslide? Based on your figure, the igneous rock is not related to a source area (dyke or sill)? Is this material displaced from upslope? If so, from where? A detailed geological map is really crucial to understand the geological context. If there is an large igneous block present in the landslide body (as shown on Figure 3), I would guess that this impermeable igneous block would largely control water fluxes and piezometric levels within the landslide.
(5) The EC data do not contribute to the interpretation of the landslide dynamics. First of all, the period of measurement is much smaller than the period of analyses of landslide dynamics. Second, EC values can vary based on the EC of the input (rainfall), the amount of evapotranspiration, and the interaction between water and soil, regolith and rocks. The data only cover surficial water bodies and open (?) drains, and do not show any difference.
It would have been interesting to have water chemistry at greater depth to see if there differences between the surgical water bodies, drains and the deeper water circulation.
In the discussion, the authors use the EC values of the drains to argue for the fact that the drains did not reach deep groundwater systems, but drain perched water bodies fed by rainfall. Do you have an idea of water transit times and water fluxes? If the water stays long enough in the perched water bodies, wouldn’t you expect a rise in the EC values?
How can you conclude that the EC values are “similar to EC values of rainfall”? Are there measurements of EC in the rain water? Can you provide these values?
(6) In the graphs in the Appendices, one can see that there is a clear difference in the data collected before mid-2016 and after mid-2016 for a selected number of piezometers and drains. Some of the piezometers have much lower values - PEG3, PP2B, PI3, PI11, and some of the drains have higher values - D24, D41. What is the reason for the differences?
Minor comments.
L297: The coordinates need to be given in LatLong. This is easier for the readers to have an idea of the location of the area without having to transform between coordinate systems.
L299 : The geology of Ecuador is highly complex. Morpho-structurally, it is divided into lithic tectonic strips, which are classified based on regional tectonic features, as well as the lithology and environments of their formation [29].
=> Sentence needs to be rewritten. What do you mean with the lithic tectonic strips, and the classification? Do you mean geological mapping? Also, why not giving the reference to the latest geological map of the region (see work by Spikings, 2001 and following)?
L308-311 : This description of vegetation-climate is too general. Referencing is inadequate - the authors refer to a geotechnical study when describing the vegetation types. One would expect that the authors look int ecosystem characteristics based on peer-reviewed literature on vegetation types in the tropical Andes. Also, when giving rainfall amounts, you need to give the name of the rainfall station, the period of measurement, and mean and st. dev. Also, when making reference to “wettest” and “somewhat drier” season, you need to give the percentage of rainfall that falls in these periods
L403: How do you collect daily evaporation data? What is the sensor that was used? Also, you say a few sentences later that you estimate ET? This is confusing. If you used Penman-Monteith, you estimated ETo - and there is no collection of evaporation data…
L409: What is the depth that the drains reach?
Figures: Please give the coordinates in LatLong
L409: What is the depth that the drains reach? please add this information to the text, as it is important to understand the link between rainfall, drain discharge and groundwater levels.
L540: Can you provide a geological map of the wider region (same area as shown in Fig 2)? Where is the igneous block coming from? What is the link with the Amaluza pluton? Where is the pluton located?
Figures: Please give the coordinates in LatLong
Figure 9: What is the depth of the drains? Is the igneous block found at the slope convexity? Isn’t there an effect of having an impermeable igneous block on water flow and fluxes in the colluvium?
Appendix C: Having a map of the residuals would be helpful to see if there are spatial patterns
Also, please check that the reference is supporting your statement. Just some examples here that need to be corrected :
It is a highly landslide-prone region, with landslides affecting the Ecuadorian landscape, society, and economy [14,15].
The references are inappropriate: (14) refers to geotechnical work by Basabe on the Middle part of the Rio Paute basin - and does not make any inference of the landslides outside this region, nor to society or economy. (15) is landslide susceptibility map of Ecuador. You need an appropriate reference from the peer-reviewed literature.
Landslides are concentrated mainly in the Andean and sub-Andean regions [16].
Here, (16) is a report on a part of the stability of part of the Guarumales landslides, and is not a comprehensive study of landslides in the Andean and sub-Andean regions. You need a reference from the scientific literature on the geomorphology of the region.
triggering factors include rainfall and seismic activity [16]…
Here (16) is about part of the Guarumaes landslide, while you are describing in this paragraph the predisposing factors of landslide occurence in the Andes mountains. Here, you need a reference to the international literature.
Round 3
Reviewer 3 Report
Just to mention that this is the 3rd version of the manuscript that I'm revising. The authors have revised some parts of the text, and have added new information in Tables and Figures. There remains several issues in the text that need to be addressed.
Main issue.
In my previous reviews, I mentioned that the geology of the site needs to be revised. The authors have revised and improved the text. I am somehow surprised that they stick to the older interpretations of Litherland, when newer maps and interpretations have been published. I would highly recommend that the description of the geology is proof-readed/carefully checked. You will see my comments below. A few notes here already: The formation of the Andes started earlier than the Mid Miocene. The Eastern Cordillera is not only composed of Paleozoic and older rocks (this is already evident from the geological map).
I would suggest that the authors carefully proofread their text, and check the tables and figures for final publication.
Other comments below:
L61-L62 : sub-tropical areas, and you give as example Europe and USA. As far as I understand, the sub-tropical regions are NOT located in Europe or USA...
L61-L65: This sentence needs to be rewritten. It is just not correct. "The underlying factors that are permanently present in the tropics...tectonic activity, and erosion processes". This might be the case for tropical mountain regions, but not for the tropics in general. Please check your text.
L65: "pose as essential factors for landslide occurrence." Not clear what you mean here. Please check your writing. Not all areas in the tropics are affected by landslides.
L68-69: You write "In Ecuador, these factors play an important role in slope stability." As stated in the previous reviews, when you write a statement like this, you need to indicate references that support your statement.
L70-78: The geology needs more attention. Please read and check the recent literature. What do you mean with "the formation of the mountain range started in the Middle Miocene"? The Eastern and Western Cordillera have different geological history, the Eastern Cordillera is much older than Miocene...
What about the "InterAdean Depression"? What do you mean with this term? Can you clarify?
Also, what do you mean with "uplift of elongated swell". Also, you state here that the Eastern Cordillera is made of Palaeozoic and older rocks - please check, also with your geological maps (!)
L76-77: In a scientific paper, you need to be more specific when you make statements. What do you mean with "landslides carry a large amount of material along slopes,... to the point of changing the Ecuadorian landscape when they are large enough"? Then, you refer to a very general paper by Guthrie and Evans. This kind of analyses has been done for the Ecuadorian Andes - see work by Guns and Vanacker (2014) on the landslide magnitude-frequency distributions from your region. See also the work by F. Sarmiento (2009) on geomorphology of Natural Hazards in Ecuador, with reference to the major landslide events.
L82: What do you mean here with the effect of "light and nutrient competition for vegetation patterns". From the sentence, I cannot see the link with landslide occurrence. Please check your statements, and look for appropriate references in the text.
L64-66: If Plio-Pleistocene volcanic deposits are covering the Ecuadorian Andes, why are they not on your geological map? Please check carefully the description of the geology.
L69: What do you mean with "structurally, it presents regional guidelines and plans of foliation"..?
L75-76: What do you mean with "anthropogenic disturbances have changed the slope scenery"? How? Where? By which processes? What is your reference for this statement?
L76: In my previous review, I asked for the time period of measurement of the rainfall data. You need to give: name of the station, period of measurement, mean and standard deviation.
L74-75: When describing the vegetation of the region, the authors refer to a book on "tropical geomorphology" and a book on "landscape transformations in the Precolumbian Americas". As stated earlier, it is important to look for appropriate references on the topic. Have a look at e.g. Jokisch and Lair (2003) or Guns (2014).
Appendix A: I would suggest that the geological map becomes one of the figures of the paper. Also, what is the source of the geological data? What about the Plio-Pleistocene deposits mentioned in the description of the study area?
Reference list
As stated in my previous reviews, the number of references to grey literature remains very high. I still note that about 40% of all references refer to material that is not published. So, this is material that is not disclosed for the reader of the paper.
References
Guns, M, Vanacker, V. (2014) Shifts in landslide frequency–area distribution after forest conversion in the tropical Andes, Anthropocene, Volume 6, 75-85.
Jokisch, B, Lair, B (2003) One last stand? Forests and change on Ecuador’s eastern cordillera. Geographical Review 92(2): 235–256.
Sarmiento, FO (2009). Geomorphology of Natural Hazards and Human-induced Disasters in Ecuador. In: Developments in Earth Surface Processes edited by E.M. Latrubesse, Elsevier, 13,149-163.
